# Revisiting a Design Choice in Gradient Temporal Difference Learning

**Xiaochi Qian**
Department of Computer Science
University of Oxford
`xiaochi.joe.qian@gmail.com`

**Shangtong Zhang**
Department of Computer Science
University of Virginia
`shangtong@virginia.edu`

## Abstract

Off-policy learning enables a reinforcement learning (RL) agent to reason counterfactually about policies that are not executed and is one of the most important ideas in RL. It, however, can lead to instability when combined with function approximation and bootstrapping, two arguably indispensable ingredients for large-scale reinforcement learning. This is the notorious deadly triad. The seminal work Sutton et al. (2008) pioneers Gradient Temporal Difference learning (GTD) as the first solution to the deadly triad, which has enjoyed massive success thereafter. During the derivation of GTD, some intermediate algorithm, called $A^\top$TD, was invented but soon deemed inferior. In this paper, we revisit this $A^\top$TD and prove that a variant of $A^\top$TD, called $A_t^\top$TD, is also an effective solution to the deadly triad. Furthermore, this $A_t^\top$TD only needs one set of parameters and one learning rate. By contrast, GTD has two sets of parameters and two learning rates, making it hard to tune in practice. We provide asymptotic analysis for $A_t^\top$TD and finite sample analysis for a variant of $A_t^\top$TD that additionally involves a projection operator. The convergence rate of this variant is on par with the canonical on-policy temporal difference learning.

## 1 Introduction

Off-policy learning (Watkins, 1989; Precup et al., 2000; Maei, 2011; Sutton et al., 2016; Li, 2019) is arguably one of the most important ideas in reinforcement learning (RL, Sutton & Barto (2018)). Different from on-policy learning (Sutton, 1988), where an RL agent learns quantities of interest of a policy by executing the policy itself, an off-policy RL agent learns quantities of interest of a policy by executing a different policy. This flexibility offers additional safety (Dulac-Arnold et al., 2019) and data efficiency (Lin, 1992; Sutton et al., 2011).

Off-policy learning, however, can lead to instability if combined with function approximation and bootstrapping, two other arguably indispensable techniques for any large-scale RL applications. The idea of function approximation (Sutton, 1988) is to represent quantities of interest with parameterized functions instead of look-up tables. The idea of bootstrapping (Sutton, 1988) is to construct update targets for an estimator by using the estimator itself recursively. This instability resulting from off-policy learning, function approximation, and bootstrapping is called the deadly triad (Baird, 1995; Sutton & Barto, 2018; Zhang, 2022).

The seminal work Sutton et al. (2008) pioneers the first solution to the deadly triad, called Gradient Temporal Difference learning (GTD). Thereafter, GTD has been massively studied and enjoyed celebrated success (Sutton et al., 2008; 2009; Maei et al., 2009; Maei & Sutton, 2010; Maei et al., 2010; Maei, 2011; Mahadevan et al., 2014; Liu et al., 2015; Du et al., 2017; Wang et al., 2017; Yu, 2017; Xu et al., 2019; Wang & Zou, 2020; Wai et al., 2020; Ghiassian et al., 2020; Zhang et al., 2021a). During the derivation of GTD in Sutton et al. (2008), an intermediate algorithm called $A^\top$TD was invented but soon deemed inferior. In Sutton et al. (2008), it is said that "although we find this algorithm interesting, we do not consider it further here because it requires $\mathcal{O}(K^2)$ memory and computation per time step". Here, $K$ refers to the feature dimension in linear function approximation. In this paper, we propose a variant of $A^\top$TD, called $A_t^\top$TD, which has $\mathcal{O}(K)$ computation per step, and the memory cost is $\mathcal{O}(K \ln^2 t)$. Here, $t$ refers to the time step. Admittedly, $\ln^2 t$ diverges

to $\infty$ eventually. However, we argue that this memory cost is negligible in any empirical implementations. For example, our universe has an age of around 14 billion years. Consider a modern 3 GHz CPU. Suppose that an RL agent runs 1 step every CPU clock and starts from the very beginning of our universe. Then until now it roughly has run $T = 14 \times 10^9 \times 3.1536 \times 10^7 \times 3 \times 10^9 \approx 10^{27}$ steps. We then have $\ln^2 T \approx 4000$. In light of this, we claim that $A_t^\top \mathrm{TD}$ does not have any real drawback in terms of memory compared with GTD. $A_t^\top \mathrm{TD}$, however, has only one set of parameters and one learning rate. By contrast, GTD has two sets of parameters and two learning rates, making it hard to tune in practice (Sutton et al., 2008). We prove that $A_t^\top \mathrm{TD}$ eventually converges to the same solution as GTD. We also demonstrate that if an additional projection operator is used, $A_t^\top \mathrm{TD}$ also enjoys the same convergence rate as the canonical on-policy TD. The assumptions in our analysis are all standard.

## 2 BACKGROUND

In this paper, all vectors are columns. We use $\|\cdot\|$ to denote the $\ell_2$ vector and matrix norm. We use functions and vectors interchangeably when it does not confuse. For example, if $f$ is a function from $\mathcal{S}$ to $\mathbb{R}$, we also use $f$ to denote a vector in $\mathbb{R}^{|\mathcal{S}|}$, whose $s$-th element is $f(s)$.

We consider an infinite horizon Markov Decision Process (MDP, Puterman (2014)) with a finite state space $\mathcal{S}$, a finite action space $\mathcal{A}$, a reward function $r : \mathcal{S} \times \mathcal{A} \to \mathbb{R}$, a transition function $p : \mathcal{S} \times \mathcal{S} \times \mathcal{A} \to [0, 1]$, and a discount factor $\gamma \in [0, 1)$. At time step 0, a state $S_0$ is sampled from some initial distribution $p_0$. At time step $t$, an agent at a state $S_t$ takes an action $A_t \sim \pi(\cdot|S_t)$. Here $\pi : \mathcal{A} \times \mathcal{S} \to [0, 1]$ is the policy being followed. A reward $R_{t+1} \doteq r(S_t, A_t)$ is then emitted, and a successor state $S_{t+1}$ is sampled from $p(\cdot|S_t, A_t)$.

The return at time step $t$ is defined as $G_t \doteq \sum_{i=0}^\infty \gamma^i R_{t+i+1}$, which allows us to define the state value function as $v_\pi(s) \doteq \mathbb{E}_{\pi,p}[G_t|S_t = s]$. The value function $v_\pi$ is the unique fixed point of the Bellman operator $\mathcal{T}_\pi v \doteq r_\pi + \gamma P_\pi v$. Here $r_\pi \in \mathbb{R}^{|\mathcal{S}|}$ is the reward vector induced by $\pi$, defined as $r_\pi(s) \doteq \sum_a \pi(a|s) r(s, a)$. $P_\pi \in \mathbb{R}^{|\mathcal{S}| \times |\mathcal{S}|}$ is the transition matrix induced by $\pi$, i.e., $P_\pi(s, s') \doteq \sum_a \pi(a|s) p(s'|s, a)$.

Estimating $v_\pi$ is one of the most important tasks in RL and is called policy evaluation. Linear function approximation is commonly used for policy evaluation (Sutton, 1988). Consider a feature function $x : \mathcal{S} \to \mathbb{R}^K$ that maps a state $s$ to a $K$-dimensional feature $x(s)$. We then use $x(s)^\top w$ to approximate $v_\pi(s)$. Here $w \in \mathbb{R}^K$ is the learnable weight. Let $X \in \mathbb{R}^{|\mathcal{S}| \times K}$ be the feature matrix, whose $s$-th row is $x(s)^\top$. The goal is then to adapt $w$ such that $Xw \approx v_\pi$. Linear TD (Sutton, 1988) updates $w$ iteratively as

$$w_{t+1} \doteq w_t + \alpha_t \left( R_{t+1} + \gamma x_{t+1}^\top w_t - x_t^\top w_t \right) x_t. \tag{1}$$

Here, we use $x_t \doteq x(S_t)$ as shorthand. Under mild conditions, the iterates $\{w_t\}$ in (1) converge almost surely (Tsitsiklis & Roy, 1996).

It is commonly the case that we want to estimate $v_\pi$ without actually sampling actions from $\pi$ due to various concerns, e.g., safety (Dulac-Arnold et al., 2019), data efficiency (Lin, 1992; Sutton et al., 2011). Off-policy learning makes this possible. In particular, instead of sampling $A_t$ according to $\pi(\cdot|S_t)$, off-policy method samples $A_t$ according to another policy $\mu$. Here, the policy $\pi$ is called the target policy and the policy $\mu$ is called the behavior policy. *For the rest of the paper, we always consider the off-policy setting*, i.e.,

$$A_t \sim \mu(\cdot|S_t), R_{t+1} = r(S_t, A_t), S_{t+1} \sim p(\cdot|S_t, A_t). \tag{2}$$

Since the behavior policy $\mu$ is different from the target policy $\pi$, importance sampling ratio is used to account for this discrepancy, which is defined as $\rho(s, a) \doteq \frac{\pi(a|s)}{\mu(a|s)}$. In particular, we use as shorthand $\rho_t \doteq \rho(S_t, A_t)$. Off-policy linear TD then updates $w$ iteratively as

$$w_{t+1} \doteq w_t + \alpha_t \rho_t \left( R_{t+1} + \gamma x_{t+1}^\top w_t - x_t^\top w_t \right) x_t. \tag{3}$$

It is well-known (Sutton et al., 2008) that if off-policy linear TD converged, it would converge to a $w_*$ satisfying

$$Aw_* + b = 0, \tag{4}$$

where

$$A \doteq X^\top D_\mu(\gamma P_\pi - I)X, \; b \doteq X^\top D_\mu r_\pi. \tag{5}$$

Here, $d_\mu$ is the stationary distribution of the Markov chain induced by the behavior policy $\mu$, and $D_\mu$ is a diagonal matrix with the diagonal being $d_\mu$. Unfortunately, the possible divergence of off-policy linear TD in (3) is well documented (Baird, 1995; Sutton et al., 2016; Sutton & Barto, 2018). This divergence exercises the deadly triad.

Instead of using off-policy linear TD in (3) to find $w_*$, one natural approach for policy evaluation in the off-policy setting is then to solve $Aw + b = 0$ directly, probably with stochastic gradient descent on the objective $L(w) \doteq \|Aw + b\|^2$. The on-policy version of this objective (i.e., with $\mu = \pi$) is first introduced in Yao & Liu (2008) to derive preconditioned TD. The off-policy version considered in this paper is first used by Sutton et al. (2008) to derive GTD, and this objective is called *the norm of the expected TD update* (NEU) in Sutton et al. (2009). The gradient of $L(w)$ can be easily computed as $\nabla L(w) = 2A^\top(Aw + b)$. One can, therefore, update $w$ as

$$w_{t+1} \doteq w_t - \alpha_t A^\top(Aw_t + b).$$

Since we do not know $A$ and $b$, we need to estimate $A^\top(Aw_t + b)$ with samples. The idea of $A^\top$TD in Sutton et al. (2008) is to estimate $A^\top$ as

$$A^\top \approx \tfrac{1}{t+1} \sum_{i=0}^{t} \rho_i(\gamma x_{i+1} - x_i)x_i^\top$$

and to estimate $Aw_t + b$ as

$$Aw_t + b \approx \rho_t\big(R_{t+1} + \gamma x_{t+1}^\top w_t - x_t^\top w_t\big)x_t.$$

As said in Sutton et al. (2008), $A^\top$TD is "essentially conventional TD(0) prefixed by an estimate of the matrix $A^\top$". Apparently, computing and store this estimate of $A^\top$ requires $\mathcal{O}(K^2)$ computation and memory per step, if we use a moving average implementation. And it is unclear whether this $A^\top$TD is convergent. Having deemed this $A^\top$TD inferior, Sutton et al. (2008) rewrite the gradient as

$$\nabla L(w) = A^\top(Aw + b) = A^\top X^\top D_\mu\left(\mathcal{T}_\pi(Xw) - Xw\right)$$

and use a secondary weight $\nu \in \mathbb{R}^K$ to approximate $X^\top D_\mu\left(\mathcal{T}_\pi(Xw) - Xw\right)$, yielding the following well-known GTD algorithm

$$\begin{aligned}
\delta_t &\doteq R_{t+1} + \gamma x_{t+1}^\top w_t - x_t^\top w_t,\\
\nu_{t+1} &\doteq \nu_t + \alpha_t\left(\rho_t\delta_t x_t - \nu_t\right),\\
w_{t+1} &\doteq w_t + \alpha_t\rho_t(x_t - \gamma x_{t+1})x_t^\top \nu_t.
\end{aligned} \tag{GTD}$$

The convergence and finite sample analysis of GTD is well established (Sutton et al., 2008; 2009; Liu et al., 2015; Wang et al., 2017; Yu, 2017).

## 3 $A_t^\top$TD: Revisiting the Design Choice of $A^\top$TD

In this paper, we refine the idea of $A^\top$TD via estimating $A^\top$ with a single sample at time $t + f(t)$ as

$$A^\top \approx \rho_{t+f(t)}\left(x_{t+f(t)} - \gamma x_{t+f(t)+1}\right)x_{t+f(t)}^\top,$$

where $f : \mathbb{N} \to \mathbb{N}$ is an increasing *gap function*. This yields the following novel algorithm:

$$\begin{aligned}
\delta_t &\doteq R_{t+1} + \gamma x_{t+1}^\top w_t - x_t^\top w_t,\\
w_{t+1} &\doteq w_t + \alpha_t\rho_{t+f(t)}\left(x_{t+f(t)} - \gamma x_{t+f(t)+1}\right)x_{t+f(t)}^\top \rho_t\delta_t x_t.
\end{aligned} \tag{$A_t^\top$TD}$$

We call it $A_t^\top$TD to highlight that it uses a single sample to estimate $A^\top$. In ($A_t^\top$TD), the term $\rho_{t+f(t)}\rho_t\big(R_{t+1} + \gamma x_{t+1}^\top w_t - x_t^\top w_t\big)$ is a scalar, the computational complexity of which is only $\mathcal{O}(K)$. If we compute the remaining term $\left(x_{t+f(t)} - \gamma x_{t+f(t)+1}\right)x_{t+f(t)}^\top x_t$ from right to left, the computational complexity is still $\mathcal{O}(K)$. In other words, the computational complexity of ($A_t^\top$TD)

is the same as (GTD). The price we pay here is that we cannot start ($A_t^\top$TD) until the $(f(0) + 1)$-th step, and we need to maintain a memory storing

$$x_t, \rho_t, x_{t+1}, \rho_{t+1}, \ldots x_{f(t)}, \rho_{f(t)}, x_{f(t)+1}. \tag{9}$$

The size of this memory is $\mathcal{O}(f(t))$. We will soon prove that the memory can be as small as $\Omega(\ln^2(t))$. We argue that this memory overhead is negligible in any empirical implementations. The gain is that we now do not need an additional weight vector, making the algorithm easy to use. We will have a few assumptions on the gap function $f$ shortly to facilitate the theoretical analysis. But one example could simply be

$$f(t) = \lfloor \ln^2(t+1) \rfloor,$$

where $\lfloor \cdot \rfloor$ is the floor function. In other words, the choice of the gap function is simple and does not depend on any unknown problem structure. To understand how this gap function works, we consider a case where $f(t) = 0 \forall t$. Then we are essentially estimating $A^\top$ and $Aw_t + b$ with the same sample $(x_t, R_{t+1}, x_{t+1})$. We will then for sure run into the well-known double sampling issue[1]. By using the gap function, we use the sample at time $t + f(t)$ to estimate $A^\top$ and the sample at time $t$ to estimate $Aw_t + b$. Despite that those two samples are still correlated due to the Markovian nature of the data stream (2), the increasing $f(t)$ gradually reduces the correlation. The theoretical analysis in the following two sections confirms that such a simple gap function is enough to guarantee the desired convergence to $w_*$ with a desired convergence rate.

We do note that throughout the paper we consider the canonical RL setting where only Markovian samples are available. If instead i.i.d. samples are available, addressing the aforementioned doubling sampling issue then becomes more straightforward – one can simply use two independent samples to estimate $A^\top$ and $Aw_t + b$. $A^\top$TD with i.i.d. samples are thoroughly investigated in Yao (2023) and we refer the reader to Yao (2023) for more details.

## 4 ASYMPTOTIC CONVERGENCE ANALYSIS OF $A_t^\top$TD

In this section, we provide an asymptotic convergence analysis of ($A_t^\top$TD). The major technical challenge lies in the increasing gap function. If $f(t)$ was a constant function, say $f(t) \equiv t_0$, then one could start analyzing ($A_t^\top$TD) via constructing an augmented Markov chain with states $Y_t \doteq \{S_t, A_t, \ldots, S_{t+t_0}, A_{t+t_0}, S_{t+t_0+1}\}$, evolving in a finite space $(\mathcal{S} \times \mathcal{A})^{t_0+1} \times \mathcal{S}$. Suppose the origin Markov chain $\{S_t\}$ is ergodic, this new chain $\{Y_t\}$ would also be ergodic, matching the ergodicity assumption of classical convergence results (e.g., Proposition 4.7 of Bertsekas & Tsitsiklis (1996)).[2] When $f(t)$ is increasing, the augmented chain, however, is now $Y_t \doteq \{S_t, A_t, \ldots, S_{t+f(t)}, A_{t+f(t)}, S_{t+f(t)+1}\}$ which evolves in an infinite space $\bigcup_{i=1}^\infty (\mathcal{S} \times \mathcal{A})^i \times \mathcal{S}$. Even if the original chain $\{S_t\}$ is ergodic, the new chain $\{Y_t\}$ still behaves poorly in that it never visits the same augmented state twice. This rules out the possibility of applying most, if not all, existing convergence results in the stochastic approximation community (e.g., Benveniste et al. (1990); Kushner & Yin (2003); Borkar (2009); Liu et al. (2025)). To proceed, we instead use the skeleton iterates technique introduced by Qian et al. (2024). The key idea of this skeleton iterates technique is to divide the non-negative real axis into intervals of length $\{T_m\}$ and examine the updates interval by interval. Importantly, we will require this $\{T_m\}$ to diminish, in a rate coordinated with the gap function $f(t)$ and the learning rate $\alpha_t$.

Besides the skeleton iterates technique, another important ingredient is the mixing of joint state distributions in Markov chains. Consider a general Markov chain $\{Y_t\}$. Assume the chain is ergodic and let $d_\mathcal{Y}$ denote its invariant distribution. Then the convergence theorem (see, e.g., Levin & Peres (2017)) yields $\lim_{t\to\infty} \Pr(Y_t = y) = d_\mathcal{Y}(y)$. This convergence is uniform in $y$ and is geometrically

---

[1]It is well-known that in general $\mathbb{E}[XY] \neq \mathbb{E}[X]\mathbb{E}[Y]$.

[2]We note that when $f(t) \equiv t_0$, another key assumption does not hold. To see this, let $\hat{A}_t \doteq \rho_t x_t (\gamma x_{t+1} - x_t)^\top$. If $f(t) \equiv t_0$, the expected updates would be governed by $-\mathbb{E}\left[\hat{A}_{t+t_0}^\top \hat{A}_t\right]$. However, this expectation is not equal to $-A^\top A$ due to the correlation between $\hat{A}_{t+t_0}$ and $\hat{A}_t$. So this expectation is unlikely to be negative definite. But canonical results typically require this expectation to be negative define. To summarize, if $f(t) \equiv t_0$, the ergodicity assumption on the Markov chain in the canonical results can be fulfilled but the negative definiteness of the expected update matrix cannot be fulfilled.

fast. Exploiting this convergence, we are able to prove the convergence of joint state distributions, i.e.,

$$\lim_{t \to \infty} \Pr\big(Y_t = y, Y_{t+f(t)} = y'\big) = d_{\mathcal{Y}}(y) d_{\mathcal{Y}}(y').$$

Intuitively, this means the dependence between $Y_t$ and $Y_{t+f(t)}$ diminishes as $t$ goes to infinity (cf. Lemma 7.1 in Vempala (2005)). In ($A_t^\top$TD), this means the bias resulting from the correlation of the two samples at time $t + f(t)$ and time $t$ diminishes gradually. Having introduced the two main technical ingredients in our analysis, we are now ready to formally describe our main results. We start with (standard) assumptions we make.

**Assumption 4.1** *The Markov chain induced by the behavior policy $\mu$ is finite, irreducible, and aperiodic. And $\mu$ covers $\pi$, i.e., $\forall(s,a), \pi(a|s) > 0 \implies \mu(a|s) > 0$.*

**Assumption 4.2** *The feature matrix $X$ has a full column rank. The matrix $A$ defined in (5) is nonsingular.*

Assumptions 4.1 and 4.2 are standard in the analysis of linear TD methods (see, e.g., Tsitsiklis & Roy (1996); Wang et al. (2017)).

**Assumption 4.3** *The learning rates $\{\alpha_t\}$ have the form of $\alpha_t = \frac{C_\alpha}{(t+1)^\nu}$, for some $\nu \in (\frac{2}{3}, 1]$.*

Assumption 4.3 considers learning rates of a specific form. This is mostly for ease of presentation.

**Assumption 4.4** *The gap function $f(t) : \mathbb{N} \to \mathbb{N}$ is increasing and satisfies $\forall \chi \in [0, 1)$, $\sum_{t=0}^{\infty} \chi^{f(t)} < \infty$. Moreover, there exist constants $\tau \in (0, \frac{3\nu-2}{2\nu})$ and $C_\tau > 0$ such that $\forall t$, $f(t) \leq C_\tau \alpha_t^{-\tau}$.*

Assumption 4.4 is the most "unnatural" assumption we make and prescribes how the gap function should be chosen. Intuitively, those conditions prevent the gap function from growing too fast. Despite seemingly complicated, Lemma 15 in the appendix confirms that simply setting

$$f(t) = \lfloor h(t) \ln(t + 1) \rfloor \tag{10}$$

with any non-negative increasing function $h(t)$ converging to $\infty$ as $t \to \infty$ fulfills the first condition of Assumption 4.4. Here $\lfloor x \rfloor$ is the floor function denoting the largest integer smaller than $x$. A concrete example satisfying Assumption 4.3 and 4.4 is

$$\nu = 1, h(t) = \ln(t + 1), f(t) = \lfloor \ln^2(t + 1) \rfloor, \tau = 0.1. \tag{11}$$

We are now ready to present our main results.

**Theorem 1** *Let Assumptions 4.1, 4.2, 4.3, & 4.4 hold. Then the iterates $\{w_t\}$ generated by ($A_t^\top$TD) satisfies*

$$\lim_{t \to \infty} w_t = w_* \quad a.s.,$$

*where $w_*$ is the TD fixed point defined in (4).*

**Proof** Following Qian et al. (2024), we define a sequence $\{T_m\}_{m=0,1,\dots}$ as

$$T_m = \frac{16 \max(C_\alpha, 1)}{(\eta+1)(m+1)^\eta}, \tag{12}$$

where $C_\alpha$ is defined in Assumption 4.3 and $\eta$ is some constant such that

$$\frac{1}{2(1-\tau)} < \eta < \frac{\nu}{2-\nu}. \tag{13}$$

Here $\nu$ and $\tau$ are defined in Assumption 4.3 and 4.4 respectively. Notably, despite that we follow the skeleton iterates technique in Qian et al. (2024), our analysis is more challenging than Qian et al. (2024) in that they only need to coordinate $\{T_m\}$ with the learning rate $\alpha_t$ but we need to coordinate $\{T_m\}$ with both the learning rate $\alpha_t$ and the gap function $f(t)$. As a result, Qian et al. (2024)

only require $\eta \in (\frac{1}{2}, \frac{\nu}{2-\nu})$ but we further require $\eta > \frac{1}{2(1-\tau)}$, which significantly complicates the analysis.

We now follow Qian et al. (2024) and divide the real line into intervals with approximate length $\{T_m\}$. To this end, we define a sequence $\{t_m\}$ as $t_0 \doteq 0$,

$$t_{m+1} \doteq \min\left\{k \mid \sum_{t=t_m}^{k-1} \alpha_t \geq T_m\right\}, \; m = 0, 1, \dots \tag{14}$$

For simplicity, define

$$\bar{\alpha}_m \doteq \sum_{t=t_m}^{t_{m+1}-1} \alpha_t, \; m = 0, 1, \dots$$

Now, the real line has been divided into intervals of lengths $\{\bar{\alpha}_m\}_{m=0,1,\dots}$. The following properties of this segmentation will be used repeatedly.

**Lemma 2** *For all $m \geq 0$ and $t \geq t_m$, we have $\alpha_t \leq T_m^2$.*

The proof is provided in Section C.1.

**Lemma 3** *For all $m \geq 0$, we have $\bar{\alpha}_m \leq 2T_m$.*

The proof is provided in Section C.2. We do note that the above two lemmas are analogous to Lemmas 1 & 2 of Qian et al. (2024) but the analysis is more challenging due to the requirement of $\eta > \frac{1}{2(1-\tau)}$. Following Qian et al. (2024), we now investigate the iterates $\{w_t\}$ interval by interval. Telescoping $(A_t^\top\text{TD})$ yields

$$w_{t_{m+1}} = w_{t_m} + \sum_{t=t_m}^{t_{m+1}-1} \alpha_t(-\hat{A}_{t+f(t)}^\top \hat{A}_t w_t - \hat{A}_{t+f(t)}^\top \hat{b}_t), \tag{15}$$

where we have used shorthand $\hat{A}_t \doteq \rho_t x_t (\gamma x_{t+1} - x_t)^\top$ and $\hat{b}_t \doteq \rho_t R_{t+1} x_t$. For ease of presentation, we define for all $m > 0$,

$$q_m = w_{t_m} + A^{-1}b. \tag{16}$$

Then, our goal is to show that $\{q_m\}$ converges to 0. Plugging in (16) into (15) yields

$$q_{m+1} = q_m + \sum_{t=t_m}^{k_{m+1}-1} \alpha_t(-\hat{A}_{t+f(t)}^\top \hat{A}_t w_t - \hat{A}_{t+f(t)}^\top \hat{b}_t)$$

$$= q_m + \sum_{t=t_m}^{k_{m+1}-1} \alpha_t \left[ -\hat{A}_{t+f(t)}^\top \hat{A}_t \left(w_t + A^{-1}b\right) - \hat{A}_{t+f(t)}^\top \left(\hat{b}_t - \hat{A}_t A^{-1}b\right) \right]$$

$$= q_m + g_{1,m} + g_{2,m} + g_{3,m} + g_{4,m},$$

where

$$g_{1,m} = \sum_{t=t_m}^{t_{m+1}-1} \alpha_t(-A^\top A q_m) = -\bar{\alpha}_m A^\top A q_m,$$

$$g_{2,m} = \sum_{t=t_m}^{t_{m+1}-1} \alpha_t \left(A^\top A - \mathbb{E}\left[\hat{A}_{t+f(t)}^\top \hat{A}_t | \mathcal{F}_{t_m+f(t_m)}\right]\right) q_m$$

$$- \sum_{t=t_m}^{t_{m+1}-1} \alpha_t \mathbb{E}\left[\hat{A}_{t+f(t)}^\top \left(\hat{b}_t - \hat{A}_t A^{-1}b\right) | \mathcal{F}_{t_m+f(t_m)}\right],$$

$$g_{3,m} = \sum_{t=t_m}^{t_{m+1}-1} \alpha_t \left(\mathbb{E}\left[\hat{A}_{t+f(t)}^\top \hat{A}_t | \mathcal{F}_{t_m+f(t_m)}\right] - \hat{A}_{t+f(t)}^\top \hat{A}_t\right) q_m$$

$$+ \sum_{t=t_m}^{t_{m+1}-1} \alpha_t \left(\mathbb{E}\left[\hat{A}_{t+f(t)}^\top \left(\hat{b}_t - \hat{A}_t A^{-1}b\right) | \mathcal{F}_{t_m+f(t_m)}\right] - \hat{A}_{t+f(t)}^\top \left(\hat{b}_t - \hat{A}_t A^{-1}b\right)\right),$$

$$g_{4,m} = \sum_{t=t_m}^{t_{m+1}-1} \alpha_t \hat{A}_{t+f(t)}^\top \hat{A}_t \left[q_m - \left(w_t + A^{-1}b\right)\right].$$

Here $\mathcal{F}_t$ denotes the $\sigma$-algebra until time $t$, i.e., $\mathcal{F}_t \doteq \sigma(w_0, S_0, A_0, \dots, S_{t-1}, A_{t-1}, S_t)$. We use the following lemmas to bound each term above. In Qian et al. (2024), they do not have terms like $f(t)$ (cf. $f(t) = 0$). As a result, their $w_{t+1}$ is adapted to $\mathcal{F}_t$. But in our analysis, due to the dependence on $\hat{A}_{t+f(t)}$, $w_{t+1}$ is *not* adapted to $\mathcal{F}_t$ and is only adapted to $\mathcal{F}_{t+f(t)}$. This greatly complicates the analysis, and we will repeatedly use Lemma 14 to address this challenge. Moreover, Assumption 4.2 implies that the matrix $A^\top A$ is positive definite, i.e., there exists a constant $\beta > 0$ such that for all $w$,

$$w^\top A^\top A w \geq \beta \|w\|^2. \tag{17}$$

This $\beta$ plays a key role in the following bounds. The finiteness of the MDP and Assumptions 4.1 & 4.2 ensure the existence of a constant $H < \infty$ such that

$$\sup_t \max\left\{\left\|\hat{A}_t\right\|, \left\|\hat{b}_t\right\|, \|A\|, \|b\|, \left\|\hat{A}_t A^{-1}b\right\|\right\} \leq H.$$

**Lemma 4** *If $e^{2T_m H^2} \leq 2$, then for all $t$ such that $t_m \leq t < t_{m+1}$, we have*

$$\left\| q_m - \left( w_t + A^{-1}b \right) \right\| \leq 8T_m H^2 (\|q_m\| + 1).$$

The proof is provided in Section C.3.

**Lemma 5** *If $2H^4 T_m \leq \beta$, then $\|q_m + g_{1,m}\|^2 \leq (1 - \beta T_m)\|q_m\|^2$.*

The proof is provided in Section C.4.

**Lemma 6** *If $T_m \leq 1$, then*

$$\|g_{2,m}\| \leq C_M T_m^{2(1-\tau)}(H+1)\left(C_\tau + L(f,\chi) + \tfrac{1}{1-\chi}\right)(\|q_m\| + 1),$$

*where $L(f,\chi) = \sum_{t=0}^{\infty} \chi^{f(t)}$, $C_\tau$ is defined in Assumption 4.4, and $C_M$ is defined in Lemma 14. Notably, $L(f, \cdot)$ is finite due to Assumption 4.4.*

The proof is provided in Section C.5.

**Lemma 7** *$\|g_{3,m}\| \leq 8T_m H^2(\|q_m\| + 1)$ and $\mathbb{E}\left[g_{3,m} | \mathcal{F}_{t_m + f(t_m)}\right] = 0$.*

The proof is provided in Section C.6.

**Lemma 8** *If $e^{2T_m H^2} \leq 2$, then $\|g_{4,m}\| \leq 8T_m^2 H^4(\|q_m\| + 1)$.*

The proof is provided in Section C.7. Putting all the bounds together, the following lemma shows that the sequence $\{\|q_m\|\}_{m \geq 0}$ is a supermartingale sequence.

**Lemma 9** *If $T_m \leq \min\left(\frac{\beta}{2H^4}, 1, \frac{\ln(2)}{2H^2}\right)$, then there exists a scalar $D$ such that*

$$\mathbb{E}\left[\|q_{m+1}\|^2 | \mathcal{F}_{t_m + f(t_m)}\right] \leq \left(1 - \beta T_m + DT_m^{2(1-\tau)}\right)\|q_m\|^2 + DT_m^{2(1-\tau)},$$

*where $\tau$ is defined in Assumption 4.4. In particular, when $DT_m^{2(1-\tau)} \leq \frac{1}{2}\beta T_m$, we have*

$$\mathbb{E}\left[\|q_{m+1}\|^2 | \mathcal{F}_{t_m + f(t_m)}\right] \leq \left(1 - \tfrac{1}{2}\beta T_m\right)\|q_m\|^2 + DT_m^{2(1-\tau)}. \tag{18}$$

The proof is provided in Section C.8. The supermartingale convergence theorem can then take over to show the convergence of $\{q_m\}$.

**Lemma 10** *$\lim_{m \to \infty} \|q_m\| = 0$ a.s.*

The proof is provided in Section C.9. With all the established lemmas, we can draw our final conclusion using Lemma 4. Since both $T_m$ and $q_m$ converges to 0 almost surely, the difference between $w_t + A^{-1}b$ and $q_m$ converges to 0 almost surely. As a result, we can conclude that $\left\{w_t + A^{-1}b\right\}_{t=0,1,\ldots}$ converges to 0, i.e., $\{w_t\}$ converges to $-A^{-1}b$ almost surely, which completes the proof. ∎

## 5 Finite Sample Analysis of $A_t^\top \text{TD}$

Theorem 1 proves the asymptotic convergence of ($A_t^\top \text{TD}$). The price we pay is a memory of size $\Omega\left(\ln^2 t\right)$ (cf. (9)). If the convergence is fast, e.g., $\mathcal{O}\left(\frac{1}{t}\right)$, the memory increases reasonably slowly, and we argue that the memory overhead is acceptable. If, however, the convergence is too slow, the memory may still become too large. To make sure that ($A_t^\top \text{TD}$) is a practical algorithm, therefore, requires performing a finite sample analysis. To this end, we, in this section, provide a finite sample

analysis of a variant of ($A_t^\top$TD), which adopts an additional projection operator and updates $\{w_t\}$ iteratively as

$$\delta_t \doteq R_{t+1} + \gamma x_{t+1}^\top w_t - x_t^\top w_t \tag{19}$$

$$w_{t+1} \doteq \Gamma\left(w_t + \alpha_t \rho_{t+f(t)}\left(x_{t+f(t)} - \gamma x_{t+f(t)+1}\right)x_{t+f(t)}^\top \rho_t \delta_t x_t\right),$$

where $\Gamma : \mathbb{R}^K \to \mathbb{R}^K$ is a projection operator onto a ball of a radius $B$. The update (19) differs from ($A_t^\top$TD) only in that it adopts an additional projection operator $\Gamma$. We, therefore, call it Projected $A_t^\top$TD. We show that the convergence rate of our Projected $A_t^\top$TD is on par with the convergence rate of the canonical on-policy linear TD in Bhandari et al. (2018), up to a few logarithm terms. Notably, adding a projection operator is a common practice in finite sample analysis of TD algorithms (see, e.g., Liu et al. (2015); Wang et al. (2017); Bhandari et al. (2018); Zou et al. (2019)) to simplify the presentation. Techniques from Srikant & Ying (2019) can indeed be used to perform finite sample analysis of the original ($A_t^\top$TD). This, however, complicates the presentation, and we, therefore, leave it for future work. We now present our main results.

**Theorem 11** *Let $B$ be large enough such that $\|w_*\| \le B$. Consider learning rates in the form of $\alpha_t = \frac{C_\alpha}{t+1}$. Let Assumptions 4.1, 4.2, & 4.4 hold. Then there exists a constant $C_0$ such that as long as $C_\alpha \ge C_0$, the iterates $\{w_t\}$ generated by Projected Direct GTD (19) satisfy*

$$\mathbb{E}\left[\|w_t - w_*\|^2\right] = \mathcal{O}\left(\frac{f(t)\ln(t)}{t}\right).$$

The proof of Theorem 11 is provided in Section A. Theorem 11, together with (10), confirms that the convergence rate of Projected $A_t^\top$TD is reasonably fast. In particular, if the configuration in (11) is used, the convergence rate of Projected $A_t^\top$TD is on-par with the on-policy linear TD (Bhandari et al., 2018) up to logarithmic factors.

## 6 EXPERIMENTS

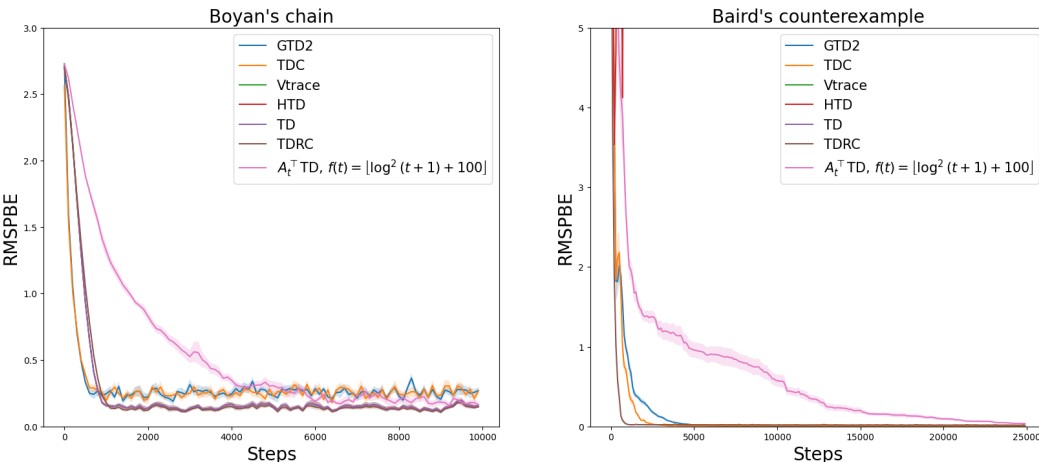

Figure 1: Comparison of ($A_t^\top$TD) with previous TD algorithms. All curves are averaged over 10 random seeds with shaded regions showing standard errors. The curves of Vtrace and HTD are invisible in Boyan's chain because they reduce to TD in the on-policy setting. The curves of Vtrace, HTD, and TD are almost invisible in Baird's counterexample because they diverge very quickly.

We now empirically compare ($A_t^\top$TD) with a few other TD algorithms with linear function approximation, including (naive) off-policy TD, GTD2 (Sutton et al., 2009), TDC (Sutton et al., 2009), Vtrace (Espeholt et al., 2018), HTD (White & White, 2016), and TDRC (Ghiassian et al., 2020). Those baselines are also used in Ghiassian et al. (2020). We consider two benchmark tasks, Boyan's chain (Boyan, 2002) and Baird's counterexample (Baird, 1995), which are also used in Ghiassian et al. (2020). Notably, Boyan's chain is an on-policy problem while Baird's counterexample is an

off-policy problem. Following Ghiassian et al. (2020), we report the square root of the mean squared projected Bellman error (RMSPBE) at each time step. We base our implementation on the open-sourced implementation from Ghiassian et al. (2020). So we refer the reader to Ghiassian et al. (2020) for details of the baselines and the tasks, as well as the exact definition of RMSPBE. For each algorithm, we tune its learning rate in $\left\{2^{-20}, \ldots, 2^{-1}, 1\right\}$ and report the results with the best learning rate (in terms of minimizing RMSPBE at the last step). As can be seen in Figure 1, in the on-policy setting, $(A_t^\top \text{TD})$ outperforms both GTD2 and TDC in terms of the final performance. In the off-policy setting, $(A_t^\top \text{TD})$ remains convergent and achieves the same final performance as GTD2, TDC, and TDRC.

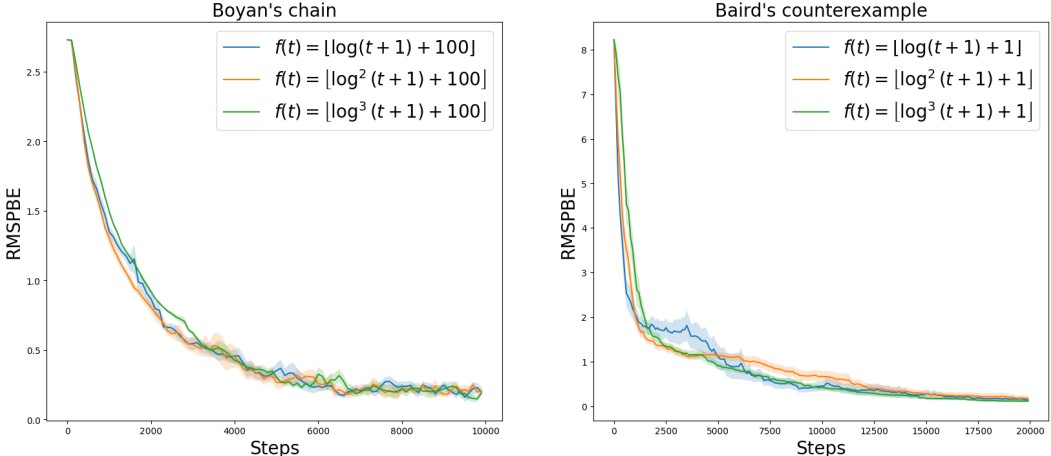

Figure 2: $(A_t^\top \text{TD})$ with different gap functions. All curves are averaged over 10 random seeds with shaded regions showing standard errors.

We also investigate different choices of the gap function. As can be seen in Figure 2, even when we set the gap function to $f(t) = \mathcal{O}(\log t)$, which does not respect Assumption 4.4, $(A_t^\top \text{TD})$ remains convergent and achieves a similar performance. This suggests that Assumption 4.4 might be overly conservative. Pushing Assumption 4.4 to its limit and finding the slowest gap function is an interesting and rewarding future work. We conjecture that if certain knowledge of the transition function of the MDP can be incorporated into the design of the gap function, the gap function can be greatly slowed down.

## 7  RELATED WORK

It is worth mentioning that the memory in $A_t^\top \text{TD}$ is conceptually different from the buffer for experience replay (Lin, 1992; Mnih et al., 2015), though both store previous transitions. A replay buffer is typically used to sample mini-batch data randomly. But $A_t^\top \text{TD}$ only deterministically uses the first and the last entries in the memory.

Instead of minimizing NEU, Feng et al. (2019) propose to minimize a kernel loss based on the Bellman error (cf. Baird (1995)). In the Markovian setting we consider in this paper, Feng et al. (2019) develop a gradient estimator of this kernel loss. This gradient estimator is consistent as the size of the replay buffer grows to infinity. However, other than the consistency, no convergence analysis is provided. Indeed, as demonstrated in Section 4, analyzing the almost sure convergence of such algorithms is extremely challenging, and no standard stochastic approximation tools apply. We conjecture that our new analysis techniques in Section 4 could further help build an almost sure convergence of their algorithms under certain kernels with some moderate regularization.

GTD is one of the many possible methods to address the deadly triad issue. Other methods include emphatic TD methods (Mahmood et al., 2015; Sutton et al., 2016; Hallak et al., 2016; Zhang & Whiteson, 2022; He et al., 2023), target networks (Zhang et al., 2021b; Fellows et al., 2023; Che et al., 2024), and density ratio methods (Hallak & Mannor, 2017; Liu et al., 2018; Nachum et al.,

2019; Zhang et al., 2020). We refer the reader to Ghiassian & Sutton (2021); Ghiassian et al. (2024) for more thorough empirical study of off-policy prediction algorithms.

## 8 CONCLUSION

In this paper, we revisit the derivation of the seminal GTD algorithm. We demonstrate that the idea behind the $A^\top$TD algorithm can lead to a new off-policy policy evaluation algorithm $A_t^\top$TD that is as competitive as GTD in terms of both asymptotic convergence, convergence rate, and per-step computation cost. $A_t^\top$TD does incur additional memory cost, which we argue is negligible in any empirical implementations. The main advantage of $A_t^\top$TD over GTD is that it has only one set of parameters and one learning rate. It is well documented that the two learning rates in GTD are hard to tune in many empirical problems. As said in Sutton et al. (2008), "we are still exploring different ways of setting the step-size parameters" (of GTD). It is worth noting again that the main contribution of this work is the rediscovery of the $A^\top$TD idea, leading to the $A_t^\top$TD algorithm. That being said, the empirical study in this work is only preliminary and we leave a more thorough empirical study for future work.

## ACKNOWLEDGEMENTS

This work is supported in part by the US National Science Foundation (NSF) under grants III-2128019 and SLES-2331904.

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

## A   PROOF OF THEOREM 11

**Proof** We first define a few shorthands. We use $\bar{g}(w)$ and $g_t(w)$ to denote the true gradient and its stochastic estimate at time $t$, respectively, i.e.,

$$\bar{g}(w) = - A^\top(Aw + b),$$
$$g_t(w) = - \hat{A}_{t+f(t)}^\top(\hat{A}_t w + \hat{b}_t).$$

We further define

$$\Lambda_t(w) = \langle w - w_*, g_t(w) - \bar{g}(w) \rangle,$$

where we recall that $w^*$ is defined in (4). The following lemma states several useful properties of the functions defined above.

**Lemma 12** *There exist constants $C_g$ and $C_{Lip}$ such that for all $w$, $w'$ with $\|w\|, \|w'\| \leq B$, we have*

$$\max\{g_t(w), \bar{g}(w), \Lambda_t(w)\} \leq C_g,$$
$$\langle w - w', \bar{g}(w) - \bar{g}(w') \rangle \leq -\beta \|w - w'\|^2,$$
$$\max\{\|g_t(w) - g_t(w')\|, \|\bar{g}(w) - \bar{g}(w')\|, |\Lambda_t(w) - \Lambda_t(w')|\} \leq C_{Lip}\|w - w'\|.$$

*We recall that $\beta$ is defined in (17).*

The proof is provided in section C.10.

**Lemma 13** *For all time $t$ and $0 < k < t$, we have the following bound*

$$\|\mathbb{E}[\Lambda_t(w_t)]\| \leq C_g C_\alpha C_{Lip} \ln\left(\frac{t}{t-k}\right) + 2B(B+1)C_M \begin{cases} 1 & k < f(t-k) \\ \chi^{f(t)} + \chi^{t-(t-k+f(t-k))} & k \geq f(t-k) \end{cases}.$$

The proof is provided in section C.11.

We are now ready to decompose the error as

$$\mathbb{E}\left[\|w_{t+1} - w_*\|^2\right]$$
$$\leq \mathbb{E}\left[\|\Gamma(w_t + \alpha_t g_t(w_t)) - w_*\|^2\right]$$
$$\leq \mathbb{E}\left[\|\Gamma(w_t + \alpha_t g_t(w_t)) - \Gamma(w_*)\|^2\right]$$
$$\leq \mathbb{E}\left[\|w_t + \alpha_t g_t(w_t) - w_*\|^2\right]$$
$$= \mathbb{E}\left[\|w_t - w_*\|^2 + \alpha_t^2\|g_t(w_t)\|^2 + 2\alpha_t\langle w_t - w_*, g_t(w_t)\rangle\right].$$

Since $\bar{g}(w_*) = 0$, we have

$$\langle w_t - w_*, g_t(w_t)\rangle$$
$$= \langle w_t - w_*, g_t(w_t) - \bar{g}(w_t) + \bar{g}(w_t) - \bar{g}(w_*)\rangle$$
$$= \Lambda_t(w_t) + \langle w_t - w_*, \bar{g}(w_t) - \bar{g}(w_*)\rangle,$$

yielding

$$\mathbb{E}\left[\|w_{t+1} - w_*\|^2\right]$$
$$\leq \mathbb{E}\left[\|w_t - w_*\|^2 + \alpha_t^2\|g_t(w_t)\|^2 + 2\alpha_t\langle w_t - w_*, g_t(w_t)\rangle\right]$$
$$\leq \mathbb{E}\left[\|w_t - w_*\|^2 + \alpha_t^2\|g_t(w_t)\|^2 + 2\alpha_t\Lambda_t(w_t) + 2\alpha_t\langle w_t - w_*, \bar{g}(w_t) - \bar{g}(w_*)\rangle\right].$$

Applying Lemma 12, we get

$$\mathbb{E}\left[\|w_{t+1} - w_*\|^2\right]$$

$$\leq \mathbb{E}\left[\|w_t - w_*\|^2 + \alpha_t^2 C_g^2 + 2\alpha_t \Lambda_t(w_t) - 2\alpha_t \beta \|w_t - w_*\|^2\right]$$

$$= (1 - 2\alpha_t\beta)\mathbb{E}[\|w_t - w_*\|^2] + \alpha_t^2 C_g^2 + 2\alpha_t \mathbb{E}[\Lambda_t(w_t)].$$

Plugging in $\alpha_t = \frac{C_\alpha}{t+1}$ and multiplying both sides by $(t+1)$, we have

$$(t+1)\mathbb{E}\left[\|w_{t+1} - w_*\|^2\right]$$

$$\leq (t+1)\left(1 - 2\beta\frac{C_\alpha}{t+1}\right)\mathbb{E}[\|w_t - w_*\|^2] + (t+1)\left(\frac{C_\alpha}{t+1}\right)^2 C_g^2 + (t+1)\cdot 2\frac{C_\alpha}{t+1}\mathbb{E}[\Lambda_t(w_t)]$$

$$\leq (t+1-2\beta C_\alpha)\mathbb{E}[\|w_t - w_*\|^2] + \frac{C_\alpha^2 C_g^2}{t+1} + 2C_\alpha \mathbb{E}[\Lambda_t(w_t)].$$

Let $C_0 = \frac{1}{2\beta}$. Then as $C_\alpha \geq C_0$, $2\beta C_\alpha \geq 2\beta C_0 = 1$. Hence, $t + 1 - 2\beta C_\alpha \leq t$. Since $\mathbb{E}\left[\|w_{t+1} - w_*\|^2\right] \geq 0$, we have

$$(t+1)\mathbb{E}\left[\|w_{t+1} - w_*\|^2\right] \leq t\mathbb{E}\left[\|w_{t+1} - w_*\|^2\right] + \frac{C_\alpha^2 C_g^2}{t+1} + 2C_\alpha \mathbb{E}[\Lambda_t(w_t)]$$

Applying the inequality recursively,

$$T\mathbb{E}\left[\|w_T - w_*\|^2\right]$$

$$\leq \sum_{t=0}^{T-1}\left(\frac{C_\alpha^2 C_g^2}{t+1} + 2C_\alpha \mathbb{E}[\Lambda_t(w_t)]\right)$$

$$= C_\alpha^2 C_g^2 \sum_{t=1}^{T}\frac{1}{t} + 2C_\alpha \sum_{t=0}^{k}\mathbb{E}[\Lambda_t(w_t)] + 2C_\alpha \sum_{t=k+1}^{T-1}\mathbb{E}[\Lambda_t(w_t)]$$

$$\leq C_\alpha^2 C_g^2[\ln(T) + 1] + 2C_\alpha \sum_{t=0}^{k}\mathbb{E}[\Lambda_t(w_t)] + 2C_\alpha \sum_{t=k+1}^{T-1}\mathbb{E}[\Lambda_t(w_t)],$$

where the last inequality comes from the bound for harmonic numbers. Applying Lemma 12 again yields

$$T\mathbb{E}\left[\|w_T - w_*\|^2\right]$$

$$\leq C_\alpha^2 C_g^2[\ln(T) + 1] + 2C_\alpha \sum_{t=0}^{k}C_g + 2C_\alpha \sum_{t=k+1}^{T-1}\mathbb{E}[\Lambda_t(w_t)]$$

$$= C_\alpha^2 C_g^2[\ln(T) + 1] + 2kC_\alpha C_g + 2C_\alpha \sum_{t=k+1}^{T-1}\mathbb{E}[\Lambda_t(w_t)]. \tag{20}$$

Now, we will use Lemma 13 to bound the last summation. Firstly, we take

$$k \doteq 2f(T).$$

Secondly, Assumption 4.4 suggests that for some $\tau \in (0, \frac{3\nu-2}{\nu})$,

$$f(t) \leq C_\tau \alpha_t^{-\tau} = C_\tau \left(\frac{C_\alpha}{(t+1)^\nu}\right)^{-\tau} = \frac{C_\tau(t+1)^{\tau\nu}}{C_\alpha^\tau}.$$

Since $\nu \leq 1$, $\tau\nu < \frac{3\nu-2}{\nu} = 3\nu - 2 \leq 1$. Hence, there exists a constant $T_0$ such that for all $T \geq T_0$, we have

$$2f(T) \leq \frac{2C_\tau(T+1)^{\tau\nu}}{C_\alpha^\tau} < T.$$

For the rest of the argument, we will assume that $T \geq T_0$, and we will then have $2f(T) < T$.

As $f$ is increasing, for all $t \in (k, T)$, we have

$$k = 2f(T) \geq 2f(t) \geq f(t-k).$$

So for $t > k$,

$$\mathbb{E}[\Lambda_t(w_t)] \leq C_g C_\alpha C_{Lip} \ln\left(\frac{t}{t-k}\right) + 2B(B+1)C_M\left(\chi^{f(t)} + \chi^{k-f(t-k)}\right) \qquad \text{(Lemma 13)}$$

$$\leq C_g C_\alpha C_{Lip} \ln\left(\frac{t}{t-k}\right) + 4B(B+1)C_M\chi^{f(t)},$$

where the last inequality uses the fact that for all $t \in (k, T)$,

$$k - f(t-k) = 2f(T) - f(t-k) > 2f(t) - f(t) = f(t).$$

Summing them up then yields

$$\sum_{t=k+1}^{T-1} \mathbb{E}[\Lambda_t(t)]$$

$$\leq \sum_{t=k+1}^{T-1} \left( C_g C_\alpha C_{Lip} \ln\left(\frac{t}{t-k}\right) + 4B(B+1)C_M\chi^{f(t)} \right)$$

$$\leq C_g C_\alpha C_{Lip} \sum_{t=k+1}^{T-1} (\ln(t) - \ln(t-k)) + 4B(B+1)C_M \sum_{t=k+1}^{T} \chi^{f(t)}$$

$$\leq C_g C_\alpha C_{Lip} \left( \sum_{t=k+1}^{T-1} \ln(t) - \sum_{t=k+1}^{T-1} \ln(t-k) \right) + 4B(B+1)C_M \sum_{t=0}^{\infty} \chi^{f(t)}$$

$$\leq C_g C_\alpha C_{Lip} \left( \sum_{t=k+1}^{T-1} \ln(t) - \sum_{t=1}^{T-k-1} \ln(t) \right) + 4B(B+1)L(f,\chi)$$

$$\leq C_g C_\alpha C_{Lip} \left( \sum_{t=T-k}^{T-1} \ln(t) - \sum_{t=1}^{k} \ln(t) \right) + 4B(B+1)L(f,\chi)$$

$$\leq C_g C_\alpha C_{Lip} \left( \sum_{t=T-k}^{T-1} \ln(t) - \ln[t - (T-k-1)] \right) + 4B(B+1)L(f,\chi)$$

$$\leq C_g C_\alpha C_{Lip} \sum_{t=T-k}^{T-1} \log\left[\frac{t}{t-(T-k-1)}\right] + 4B(B+1)L(f,\chi)$$

$$\leq C_g C_\alpha C_{Lip} \sum_{t=T-k}^{T-1} \log(T) + 4B(B+1)L(f,\chi)$$

$$\leq C_g C_\alpha C_{Lip} k \log(T) + 4B(B+1)L(f,\chi), \qquad (21)$$

where second last inequality holds because

$$\log\left[\frac{t}{t-(T-k-1)}\right] \leq \log(T)$$

holds for all $T - k - 1 < t < T$ and $L(f,\chi) = \sum_{t=0}^{\infty} \chi^{f(t)} < \infty$ due to Assumption 4.4.

Plugging (21) into (20) then yields

$$T\mathbb{E}[\|w_T - w_*\|^2]$$

$$\leq C_\alpha^2 C_g^2 [\ln(T) + 1] + 2kC_\alpha C_g + 2C_\alpha[C_g C_\alpha C_{Lip} k \log(T) + 4B(B+1)L(f,\chi)]$$

$$= C_\alpha^2 C_g^2 [\ln(T) + 1] + 2kC_\alpha C_g + 2C_g C_\alpha^2 C_{Lip} k \log(T) + 8C_\alpha B(B+1)L(f,\chi),$$

Since we have defined $k \doteq 2f(T)$, we have that for $T \geq T_0$,

$$
\begin{aligned}
&\mathbb{E}[\|w_T - w_*\|^2] \\
&\leq \frac{C_\alpha^2 C_g^2 [\ln(T) + 1]}{T} + \frac{4f(T)C_\alpha C_g}{T} + \frac{4C_g C_\alpha^2 C_{Lip} f(T) \log(T)}{T} + \frac{8C_\alpha B(B+1)L(f,\chi)}{T}.
\end{aligned}
$$

Thus, for all $T$, we have

$$
\mathbb{E}[\|w_T - w_*\|^2] = \mathcal{O}\left(\frac{f(T)\ln(T)}{T}\right),
$$

which completes the proof. ∎

## B    AUXILIARY LEMMAS

**Lemma 14** *Let Assumption 4.1 hold. Then there exists a constant $C_M > 0$ and $\chi \in [0,1)$ such that*

$$
\left\| \mathbb{E}\left[\hat{A}_t\right] - A \right\| \leq C_M \chi^t, \tag{22}
$$

$$
\left\| \mathbb{E}\left[\hat{A}_{t+k}^\top \hat{A}_t\right] - \mathbb{E}\left[\hat{A}_{t+k}\right]^\top \mathbb{E}\left[\hat{A}_t\right] \right\| \leq C_M \chi^k, \tag{23}
$$

$$
\left\| \mathbb{E}\left[\hat{A}_{t+k}^\top \hat{A}_t | \mathcal{F}_l\right] - A^\top A \right\| \leq C_M \begin{cases} 1 & t < l \\ \chi^k + \chi^{t-l} & t \geq l \end{cases}. \tag{24}
$$

*Similarly,*

$$
\left\| \mathbb{E}\left[\hat{b}_t\right] - A \right\| \leq C_M \chi^t,
$$

$$
\left\| \mathbb{E}\left[\hat{A}_{t+k}^\top \hat{b}_t\right] - \mathbb{E}\left[\hat{A}_{t+k}\right]^\top \mathbb{E}\left[\hat{b}_t\right] \right\| \leq C_M \chi^k,
$$

$$
\left\| \mathbb{E}\left[\hat{A}_{t+k}^\top \hat{b}_t | \mathcal{F}_l\right] - A^\top b \right\| \leq C_M \begin{cases} 1 & t < l \\ \chi^k + \chi^{t-l} & t \geq l \end{cases}.
$$

**Proof** For the simplicity of display, we include the proof only for the first half of the lemma. The second half is identical up to change of notations and is, therefore, omitted to avoid verbatim.

Define an augmented chain $\{Y_t\}$ evolving in

$$
\mathcal{Y} \doteq \{(s, a, s') \in \mathcal{S} \times \mathcal{A} \times \mathcal{S} \mid d_\mu(s) > 0, \mu(a|s) > 0, p(s'|s, a) > 0\}
$$

as

$$
Y_t \doteq (S_t, A_t, S_{t+1}).
$$

According to the definition of $\mathcal{F}_t$, it can be easily seen that $Y_t$ is adapted to $\mathcal{F}_{t+1}$. Assumption 4.1 immediately ensures that $\{Y_t\}$ is also ergodic with a stationary distribution

$$
d_Y(y) = d_\mu(s)\mu(a|s)p(s'|s, a).
$$

Here we have used $y$ as shorthand for $(s, a, s')$. Define functions

$$
\begin{aligned}
\hat{A}(y) &\doteq \rho(s, a)x(s)\left(\gamma x(s') - x(s)\right)^\top, \\
\hat{b}(y) &\doteq \rho(s, a)x(s)r(s, a).
\end{aligned}
$$

It can then be easily computed that

$$
\begin{aligned}
\hat{A}_t &= \hat{A}(Y_t), \\
A &= \sum_y d_Y(y)\hat{A}(y).
\end{aligned}
$$

Assumption 4.1 ensures that the chain $\{Y_t\}$ mixes geometrically fast. In other words, there exist constants $\chi \in [0, 1)$ and $C_0 > 0$ such that for any $t$ and $k$,

$$\max_y \sum_{y'} |\Pr(Y_{t+k} = y'|Y_t = y) - d_Y(y')| \leq C_0 \chi^k.$$

This is a well-known result, and we refer the reader to Theorem 4.7 of Levin & Peres (2017) for detailed proof. Then we have

$$\left\| \mathbb{E}\left[\hat{A}_t\right] - A \right\|$$

$$= \left\| \sum_y \Pr(Y_t = y)\hat{A}(y) - \sum_y d_Y(y)\hat{A}(y) \right\|$$

$$\leq \sum_y \|\Pr(Y_t = y) - d_Y(y)\| \left\| \hat{A}(y) \right\|$$

$$\leq \max_y \left\| \hat{A}(y) \right\| \sum_y \|\Pr(Y_t = y) - d_Y(y)\|$$

$$\leq H C_0 \chi^t,$$

which completes the proof of (22). Similarly, we have

$$\left\| \mathbb{E}\left[\hat{A}_{t+k}^\top \hat{A}_t\right] - \mathbb{E}\left[\hat{A}_{t+k}\right] \mathbb{E}\left[\hat{A}_t\right] \right\|$$

$$= \left\| \sum_{y'} \sum_y \Pr(Y_{t+k} = y', Y_t = y)\hat{A}(y')^\top \hat{A}(y) - \left( \sum_{y'} \Pr(Y_{t+k} = y')\hat{A}(y') \right)^\top \left( \sum_y \Pr(Y_t = y)\hat{A}(y) \right) \right\|$$

$$= \left\| \sum_{y'} \sum_y \left( \Pr(Y_{t+k} = y', Y_t = y) - \Pr(Y_{t+k} = y') \Pr(Y_t = y) \right) A(y')^\top \hat{A}(y) \right\|$$

$$\leq \max_{y'} \left\| \hat{A}(y')^\top \right\| \max_y \left\| \hat{A}(y) \right\| \left\| \sum_{y'} \sum_y \left( \Pr(Y_{t+k} = y', Y_t = y) - \Pr(Y_{t+k} = y') \Pr(Y_t = y) \right) \right\|$$

$$\leq H \cdot H \sum_{y'} \sum_y |\Pr(Y_{t+k} = y', Y_t = y) - \Pr(Y_{t+k} = y') \Pr(Y_t = y)|$$

$$\leq H^2 \sum_{y'} \sum_y |\Pr(Y_{t+k} = y'|Y_t = y) \Pr(Y_t = y) - \Pr(Y_{t+k} = y') \Pr(Y_t = y)|$$

$$\leq H^2 \sum_{y'} \sum_y |\Pr(Y_{t+k} = y'|Y_t = y) - \Pr(Y_{t+k} = y')| \Pr(Y_t = y)$$

$$\leq H^2 \sum_{y'} \sum_y |\Pr(Y_{t+k} = y'|Y_t = y) - \Pr(Y_{t+k} = y')|$$

$$\leq H^2 \sum_{y'} \sum_y |\Pr(Y_{t+k} = y'|Y_t = y) - d_Y(y')| + |d_Y(y') - \Pr(Y_{t+k} = y')|$$

$$\leq H^2 \sum_y \left( \sum_{y'} |\Pr(Y_{t+k} = y'|Y_t = y) - d_Y(y')| + \sum_{y'} |\Pr(Y_{t+k} = y') - d_Y(y')| \right)$$

$$\leq H^2 \sum_y \left( C_0 \chi^k + C_0 \chi^{t+k} \right)$$

$$\leq 2H^2 |\mathcal{Y}| C_0 \chi^k,$$

which proves (23). This also suggests $\forall l$,

$$\left\| \mathbb{E}\left[\hat{A}_{t+k}^\top \hat{A}_t \mid \mathcal{F}_l\right] - \mathbb{E}\left[\hat{A}_{t+k} \mid \mathcal{F}_l\right]^\top \mathbb{E}\left[\hat{A}_t \mid \mathcal{F}_l\right]\right\| \le 2H^2|\mathcal{Y}|C_0\chi^k.$$

To see this, we consider the two cases of whether $l < t$ separately.

**Case 1**: $l < t$. Then, by the Markov property,

$$\left\| \mathbb{E}\left[\hat{A}_{t+k}^\top \hat{A}_t \mid \mathcal{F}_l\right] - \mathbb{E}\left[\hat{A}_{t+k} \mid \mathcal{F}_l\right]^\top \mathbb{E}\left[\hat{A}_t \mid \mathcal{F}_l\right]\right\|$$

$$= \left\| \mathbb{E}\left[\hat{A}_{t+k}^\top \hat{A}_t \mid Y_l\right] - \mathbb{E}\left[\hat{A}_{t+k} \mid Y_l\right]^\top \mathbb{E}\left[\hat{A}_t \mid Y_l\right]\right\|$$

$$\le 2H^2|\mathcal{Y}|C_0\chi^k.$$

**Case 2**: $l \ge t$. Then $\hat{A}_t$ is deterministic given $\mathcal{F}_l$. and

$$\left\| \mathbb{E}\left[\hat{A}_{t+k}^\top \hat{A}_t \mid \mathcal{F}_l\right] - \mathbb{E}\left[\hat{A}_{t+k} \mid \mathcal{F}_l\right]^\top \mathbb{E}\left[\hat{A}_t \mid \mathcal{F}_l\right]\right\|$$

$$= \left\| \mathbb{E}\left[\hat{A}_{t+k}^\top \mathbb{E}\left[\hat{A}_t \mid \mathcal{F}_l\right] \mid \mathcal{F}_l\right] - \mathbb{E}\left[\hat{A}_{t+k} \mid \mathcal{F}_l\right]^\top \mathbb{E}\left[\hat{A}_t \mid \mathcal{F}_l\right]\right\|$$

$$= \left\| \mathbb{E}\left[\hat{A}_{t+k} \mid \mathcal{F}_l\right]^\top \mathbb{E}\left[\hat{A}_t \mid \mathcal{F}_l\right] - \mathbb{E}\left[\hat{A}_{t+k} \mid \mathcal{F}_l\right]^\top \mathbb{E}\left[\hat{A}_t \mid \mathcal{F}_l\right]\right\|$$

$$= 0.$$

Lastly, combining the geometrical convergence suggested in (22) and the geometrically decaying correlation implied by (23), we can prove (24) in the following manner. First,

$$\left\| \mathbb{E}\left[\hat{A}_{t+k}^\top \hat{A}_t | \mathcal{F}_l\right] - A^\top A\right\|$$

$$\le \left\| \mathbb{E}\left[\hat{A}_{t+k}^\top \hat{A}_t | \mathcal{F}_l\right] - \mathbb{E}\left[\hat{A}_{t+k}|\mathcal{F}_l\right]^\top \mathbb{E}\left[\hat{A}_t|\mathcal{F}_l\right] + \mathbb{E}\left[\hat{A}_{t+k}|\mathcal{F}_l\right]^\top \mathbb{E}\left[\hat{A}_t|\mathcal{F}_l\right] - A^\top A\right\|$$

$$\le \left\| \mathbb{E}\left[\hat{A}_{t+k}^\top \hat{A}_t | \mathcal{F}_l\right] - \mathbb{E}\left[\hat{A}_{t+k}|\mathcal{F}_l\right]^\top \mathbb{E}\left[\hat{A}_t|\mathcal{F}_l\right]\right\| + \left\| \mathbb{E}\left[\hat{A}_{t+k}|\mathcal{F}_l\right]^\top \mathbb{E}\left[\hat{A}_t|\mathcal{F}_l\right] - A^\top A\right\|$$

$$\le 2H^2|\mathcal{Y}|C_0\chi^k + \left\| \mathbb{E}\left[\hat{A}_{t+k}|\mathcal{F}_l\right]^\top \mathbb{E}\left[\hat{A}_t|\mathcal{F}_l\right] - \mathbb{E}\left[\hat{A}_{t+k}|\mathcal{F}_l\right]^\top A + \mathbb{E}\left[\hat{A}_{t+k}|\mathcal{F}_l\right]^\top A - A^\top A\right\|$$

$$\le 2H^2|\mathcal{Y}|C_0\chi^k + \left\| \mathbb{E}\left[\hat{A}_{t+k}|\mathcal{F}_l\right]^\top (\mathbb{E}\left[\hat{A}_t|\mathcal{F}_l\right] - A)\right\| + \left\| (\mathbb{E}\left[\hat{A}_{t+k}|\mathcal{F}_l\right] - A)^\top A\right\|$$

$$\le 2H^2|\mathcal{Y}|C_0\chi^k + \left\| \mathbb{E}\left[\hat{A}_{t+k}|\mathcal{F}_l\right]^\top\right\| \left\| \mathbb{E}\left[\hat{A}_t|\mathcal{F}_l\right] - A\right\| + \left\| (\mathbb{E}\left[\hat{A}_{t+k}|\mathcal{F}_l\right] - A)^\top\right\| \|A\|$$

$$\le 2H^2|\mathcal{Y}|C_0\chi^k + H\left\| \mathbb{E}\left[\hat{A}_t|\mathcal{F}_l\right] - A\right\| + H\left\| \mathbb{E}\left[\hat{A}_{t+k}|\mathcal{F}_l\right] - A\right\|$$

$$\le 2H^2|\mathcal{Y}|C_0\chi^k + H\left( \left\| \mathbb{E}\left[\hat{A}_t|\mathcal{F}_l\right] - A\right\| + \left\| \mathbb{E}\left[\hat{A}_{t+k}|\mathcal{F}_l\right] - A\right\|\right).$$

We now bound the last term. For $t < l$, we use the trivial bound

$$\left\| \mathbb{E}\left[\hat{A}_t|\mathcal{F}_l\right] - A\right\| + \left\| \mathbb{E}\left[\hat{A}_{t+k}|\mathcal{F}_l\right] - A\right\| \le 4H.$$

For $t \ge l$, both $Y_t$ and $Y_{t+k}$ are not adapted to $\mathcal{F}_l$. We, therefore, have

$$\left\| \mathbb{E}\left[\hat{A}_t|\mathcal{F}_l\right] - A\right\| + \left\| \mathbb{E}\left[\hat{A}_{t+k}|\mathcal{F}_l\right] - A\right\| \le HC_0\chi^{t-l} + HC_0\chi^{t+k-l} \le 2HC_0\chi^{t-l}.$$

Combining the results, we obtain

$$\left\| \mathbb{E}\left[\hat{A}_{t+k}^\top \hat{A}_t|\mathcal{F}_l\right] - A^\top A\right\| \le 2H^2|\mathcal{Y}|C_0\chi^k + \begin{cases} 4H^2 & t < l \\ 2H^2C_0\chi^{t-l} & t \ge l \end{cases}$$

$$= \begin{cases} 2H^2|\mathcal{Y}|C_0 + 4H^2 & t < l \\ 2H^2|\mathcal{Y}|C_0\chi^k + 2H^2C_0\chi^{t-l} & t \ge l \end{cases}, \qquad \text{(Since } \chi \le 1\text{)}$$

which completes the proof of (24). ∎

**Lemma 15** *If the gap function $f(t) = \lfloor h(t) \ln(t) \rfloor$ where $h$ is a non-negative, increasing function tending to infinity, then $\sum_{t=0}^{\infty} \chi^{f(t)}$ for all $\chi \in (0, 1)$.*

**Proof** Firstly, we should note that for all $t$, $\lfloor h(t) \ln(t) \rfloor > h(t) \ln(t) - 1$. Therefore, take arbitrary $\chi \in (0, 1)$,

$$\sum_{t=0}^{\infty} \chi^{f(t)} \leq \sum_{t=0}^{\infty} \chi^{h(t) \ln(t) - 1} = \frac{1}{\chi} \sum_{t=0}^{\infty} e^{\ln(\chi) h(t) \ln(t)} = \frac{1}{\chi} \sum_{t=0}^{\infty} t^{\ln(\chi) h(t)}.$$

Since $h$ is increasing in $t$ and tending to infinity, there exists a $T$ such that got all $t \geq T$, $f(t) \geq -\frac{2}{\ln(\chi)}$. Then, for all $t \geq T$, $t^{\ln(\chi) h(t)} \leq t^{-2}$. Thus, by comparison test and p-test, we can conclude that $\sum_{t=0}^{\infty} \chi^{f(t)} \leq \sum_{t=0}^{\infty} t^{-2} < \infty$. ∎

## C  PROOF OF TECHNICAL LEMMAS

### C.1  PROOF OF LEMMA 2

**Proof** We proceed via induction on $m$. In particular, we prove the following two inequalities for all $m$:

$$t_m \geq \frac{m^{\eta+1}}{16 \max(C_\alpha, 1)}, \tag{25}$$

$$\alpha_t \leq T_m^2, \forall t \geq t_m. \tag{26}$$

**Base Case m=0**: Obviously we have

$$t_0 = 0 = \frac{0^{\eta+1}}{16 \max(C_\alpha, 1)},$$

so (25) holds for $m = 0$. Since $\eta \in (0, 1]$, we have

$$T_0 = \frac{16 \max(C_\alpha, 1)}{\eta + 1} \geq 8 \max(C_\alpha, 1).$$

Hence, for all $t \geq t_0 = 0$,

$$\alpha_t = \frac{C_\alpha}{(t+1)^\nu} \leq C_\alpha \leq 8 \max(C_\alpha, 1)^2 \leq T_0^2.$$

So (26) holds.

**Induction Step**: Suppose (25) and (26) hold for $m = k$. We now verify them for $m = k + 1$. Letting $m = k$ in (26) yields

$$T_k \leq \sum_{t=t_k}^{t_{k+1}-1} \alpha_t \qquad \text{(Defintion of } \{t_m\} \text{ in (14))}$$

$$\leq \sum_{t=t_k}^{t_{k+1}-1} T_k^2 = (t_{k+1} - t_k) T_k^2.$$

Dividing both sides by $T_k$ yields

$$t_{k+1} - t_k \geq \frac{1}{T_k} = \frac{(\eta + 1)(k + 1)^\eta}{16 \max(C_\alpha, 1)}.$$

Consequently, we have

$$
\begin{aligned}
t_{k+1} \geq & t_k + \frac{(\eta+1)(k+1)^\eta}{16\max(C_\alpha,1)} \\
\geq & \frac{k^{\eta+1}}{16\max(C_\alpha,1)} + \frac{(\eta+1)(k+1)^\eta}{16\max(C_\alpha,1)},
\end{aligned}
$$

where the last inequality results from inductive hypothesis (25). Since $\frac{(\eta+1)(k+1)^\eta}{16\max\{C_\alpha,1\}}$ is monotonically increasing in $k$, we have

$$
\frac{(\eta+1)(k+1)^\eta}{16\max(C_\alpha,1)} \geq \int_k^{k+1} \frac{(\eta+1)(k+1)^\eta}{16\max(C_\alpha,1)} = \frac{(k+1)^{\eta+1}}{16\max(C_\alpha,1)} - \frac{k^{\eta+1}}{16\max(C_\alpha,1)}.
$$

We have thus verified (25) for $m = k+1$, i.e.

$$
t_{k+1} \geq \frac{k^{\eta+1}}{16\max(C_\alpha,1)} + \frac{(k+1)^{\eta+1}}{16\max(C_\alpha,1)} - \frac{k^{\eta+1}}{16\max(C_\alpha,1)} = \frac{(k+1)^{\eta+1}}{16\max(C_\alpha,1)}.
$$

To verify (26) for $m = k+1$, we will make use of our just proven (25) with $m = k+1$. Take arbitrary $t \geq t_{k+1}$. As $\alpha_t = \frac{C_\alpha}{(t+1)^\nu}$ is a monotonically decreasing in $t$, we have

$$
\alpha_t \leq \alpha_{t_{k+1}} = \frac{C_\alpha}{(t_{k+1}+1)^\nu} \leq \frac{C_\alpha}{t_{k+1}^\nu}.
$$

Using (25) with $m = k+1$, we get

$$
\alpha_t \leq \frac{C_\alpha}{t_{k+1}^\nu} \leq \frac{C_\alpha}{\left(\frac{1}{16\max(C_\alpha,1)}(k+1)^{\eta+1}\right)^\nu} \leq \frac{16^\nu C_\alpha \max(C_\alpha,1)^\nu}{(k+2)^{(\eta+1)\nu}} \left(\frac{k+2}{k+1}\right)^{(\eta+1)\nu}.
$$

As $\frac{k+2}{k+1} \leq 2$ for $k \geq 0$ and $\eta,\nu \in (0,1]$, we have

$$
\left(\frac{k+2}{k+1}\right)^{(\eta+1)\nu} \leq 2^{(\eta+1)\nu}.
$$

Moreover, since $C_\alpha \leq \max(C_\alpha,1)$, we have

$$
C_\alpha \max(C_\alpha,1)^\nu \leq \max(C_\alpha,1)^{1+\nu}.
$$

Thus we have

$$
\alpha_t \leq \frac{16^\nu C_\alpha \max(C_\alpha,1)^\nu}{(k+2)^{(\eta+1)\nu}} \left(\frac{k+2}{k+1}\right)^{(\eta+1)\nu} \leq \frac{2^{(\eta+5)\nu}\max(C_\alpha,1)^{1+\nu}}{(k+2)^{(\eta+1)\nu}}.
$$

Using $\eta,\nu \in (0,1]$, we have

$$
\begin{aligned}
0 \leq & (\eta+5)\nu \leq 6, \\
\max(C_\alpha,1)^{1+\nu} \leq & \max(C_\alpha,1)^2.
\end{aligned}
$$

The definition of $\eta$ in (13) implies

$$
2\eta < (\eta+1)\nu.
$$

Hence, we get

$$
\alpha_t \leq \frac{2^{(\eta+5)\nu}\max(C_\alpha,1)^{1+\nu}}{(k+2)^{(\eta+1)\nu}} \leq \frac{64\max(C_\alpha,1)^2}{(k+2)^{2\eta}}.
$$

The second inequality in (13) together with the fact that $\nu \in (0,1]$ implies that $\eta < 1$. Consequently, we have $(\eta+1)^2 < 4$. Therefore, $64 < \frac{256}{(1+\eta)^2}$ and

$$
\alpha_t \leq \frac{64\max(C_\alpha,1)^2}{(k+2)^{2\eta}} \leq \frac{256\max(C_\alpha,1)^2}{(\eta+1)^2(k+2)^{2\eta}} = \left(\frac{16\max(C_\alpha,1)}{(\eta+1)(k+2)^\eta}\right)^2 = T_{k+1}^2.
$$

We have now verified that (26) holds for $m = k+1$, which completes the induction. ∎

## C.2   PROOF OF LEMMA 3

**Proof** The fact that $\nu \in (0,1)$ and (13) implies

$$\eta < \frac{\nu}{2-\nu} \le \nu.$$

The fact that $\eta \in [0,1]$ implies

$$\frac{16}{\eta+1} > 8.$$

Consequently, we have

$$T_m = \frac{16}{\eta+1} \frac{\max(C_\alpha, 1)}{(m+1)^\eta} > 8 \frac{C_\alpha}{(m+1)^\nu} = 8\alpha_m.$$

The definition of $\{t_m\}$ in (14) implies that $t_{m+1} - t_m \ge 1$ for all $m \ge 0$, so we have

$$t_m \ge m.$$

Moreover, because $\alpha_t = \frac{C_\alpha}{(t+1)^\nu}$ is decreasing in $t$, for all $t \ge t_m \ge m$, we have

$$\alpha_t \le \alpha_m \le \frac{T_m}{8}.$$

The definition of $\{t_m\}$ in (14) also implies that $\sum_{t=t_m}^{t_{m+1}-2} \alpha_t < T_m$. Then we have

$$\bar{\alpha}_m = \sum_{t=t_m}^{t_{m+1}-1} \alpha_t = \sum_{t=t_m}^{t_{m+1}-2} \alpha_t + \alpha_{t_{m+1}-1} \le T_m + \frac{T_m}{8} = \frac{9T_m}{8} \le 2T_m,$$

which completes the proof. Note here we have used the convention that $\sum_{t=i}^{j} \alpha_t \doteq 0$ if $i > j$. ∎

## C.3   PROOF OF LEMMA 4

**Proof** For all $t \ge 0$,

$$
\begin{aligned}
&\left\| w_{t+1} + A^{-1}b \right\| \\
=&\left\| (w_t + A^{-1}b) + \alpha_t \hat{A}_{t+f(t)}^\top \left( \hat{A}_t(w_t + A^{-1}b) - \hat{A}_t A^{-1}b + \hat{b}_t \right) \right\| \\
\le&\left\| w_t + A^{-1}b \right\| + \alpha_t \left\| \hat{A}_{t+f(t)}^\top \right\| \left( \left\| \hat{A}_t \right\| \left\| w_t + A^{-1}b \right\| + \left\| \hat{A}_t A^{-1}b \right\| + \|\hat{b}_t\| \right) \\
\le&\left\| w_t + A^{-1}b \right\| + \alpha_t H \left( H \left\| w_t + A^{-1}b \right\| + H + H \right) \\
=&\left\| w_t + A^{-1}b \right\| + \alpha_t H^2 \left( \left\| w_t + A^{-1}b \right\| + 2 \right)
\end{aligned}
\tag{27}
$$

Therefore, by adding 2 to both sides, we get

$$\left\| w_{t+1} + A^{-1}b \right\| + 2 \le (1 + \alpha_t H^2) \left( \left\| w_t + A^{-1}b \right\| + 2 \right).$$

Applying the inequality iteratively, we have that for all $t$ satisfying $t_m \le t \le t_{m+1}$

$$\left\| w_t + A^{-1}b \right\| + 2 \le \left( \left\| w_{t_m} + A^{-1}b \right\| + 2 \right) \prod_{j=t_m}^{t_{m+1}} (1 + \alpha_t H^2) \le e^{\bar{\alpha}_m H^2} \left( \left\| w_{t_m} + A^{-1}b \right\| + 2 \right),$$

where for the last two inequalities, we used the fact that

$$\prod_{j=t_m}^{t_{m+1}-1} (1 + \alpha_t H^2) \le \exp\left( \sum_{j=t_m}^{t_{m+1}-1} \alpha_t H^2 \right) = \exp\left( \bar{\alpha}_m H^2 \right).$$

As $\bar{\alpha}_m \leq 2T_m$ (Lemma 3) and $e^{2T_m H^2} \leq 2$,

$$\begin{aligned}
\left\|w_t + A^{-1}b\right\| + 2 &\leq e^{\bar{\alpha}_m H^2}\left(\left\|w_{t_m} + A^{-1}b\right\| + 2\right) \\
&\leq e^{2T_m H^2}\left(\left\|w_{t_m} + A^{-1}b\right\| + 2\right) \\
&\leq 2\left(\left\|w_{t_m} + A^{-1}b\right\| + 2\right).
\end{aligned}$$

Hence,

$$\left\|w_t + A^{-1}b\right\| \leq 2\left(\left\|w_{t_m} + A^{-1}b\right\| + 1\right). \tag{28}$$

Therefore, for all $t_m \leq t \leq t_{m+1}$, we have

$$\begin{aligned}
&\left\|\left(w_t + A^{-1}b\right) - q_m\right\| \\
={}&\left\|\left(w_t + A^{-1}b\right) - \left(w_{t_m} + A^{-1}b\right)\right\| \\
\leq{}&\left\|\sum_{j=t_m}^{t-1}\left(w_{j+1} + A^{-1}b\right) - \left(w_j + A^{-1}b\right)\right\| \\
\leq{}&\sum_{j=t_m}^{t-1}\left\|\left(w_{j+1} + A^{-1}b\right) - \left(w_j + A^{-1}b\right)\right\| \\
={}&\sum_{j=t_m}^{t-1}\left\|\alpha_j \hat{A}_{j+f(j)}^\top\left(\hat{A}_j(w_t + A^{-1}b) - \hat{A}_j A^{-1}b + \hat{b}_j\right)\right\| \\
\leq{}&\sum_{j=t_m}^{t_{m+1}-1}\alpha_t H^2\left(\left\|w_t + A^{-1}b\right\| + 2\right) && \text{(Similar to (27))} \\
\leq{}&\sum_{j=t_m}^{t_{m+1}-1}\alpha_t H^2\left(2\left\|w_{t_m} + A^{-1}b\right\| + 4\right) && \text{(Using (28))} \\
={}&2\bar{\alpha}_m H^2\left(\left\|w_{t_m} + A^{-1}b\right\| + 2\right) \\
\leq{}&4T_m H^2\left(\left\|w_{t_m} + A^{-1}b\right\| + 2\right) && \text{(Using Lemma 3)} \\
\leq{}&8T_m H^2\left(\left\|w_{t_m} + A^{-1}b\right\| + 1\right) \\
={}&8T_m H^2\left(\|q_m\| + 1\right),
\end{aligned}$$

which completes the proof. ∎

### C.4 PROOF OF LEMMA 5

**Proof**

$$\begin{aligned}
\left\|q_m + g_{1,m}\right\|^2 = \left\|q_m - \bar{\alpha}_m A^\top A q_m\right\|^2 &= \left\|\left(I - \bar{\alpha}_m A^\top A\right)q_m\right\|^2 \\
&= q_m^\top\left(I - \bar{\alpha}_m A^\top A\right)^\top\left(I - \bar{\alpha}_m A^\top A\right)q_m \\
&= \|q_m\|^2 - 2\bar{\alpha}_m q_m^\top A^\top A q_m + \bar{\alpha}_m^2\left\|A^\top A q_m\right\|^2 \\
&\leq \|q_m\|^2 - 2\bar{\alpha}_m \beta\|q_m\|^2 + \bar{\alpha}_m^2\left\|A^\top\right\|^2\|A\|^2\|q_m\|^2 \\
&\leq \|q_m\|^2 - 2\bar{\alpha}_m \beta\|q_m\|^2 + \bar{\alpha}_m^2 H^4\|q_m\|^2 \\
&\leq \|q_m\|^2 - 2\bar{\alpha}_m \beta\|q_m\|^2 + 2\bar{\alpha}_m T_m H^4\|q_m\|^2 && \text{(Lemma 3)} \\
&\leq \|q_m\|^2 - 2\bar{\alpha}_m \beta\|q_m\|^2 + \bar{\alpha}_m \beta\|q_m\|^2 && \text{(Assumption of this Lemma)} \\
&\leq \left(1 - \beta\bar{\alpha}_m\right)\|q_m\|^2 \\
&\leq \left(1 - \beta T_m\right)\|q_m\|^2. && \text{(Definition of $T_m$ in (14))}
\end{aligned}$$

∎

## C.5 PROOF OF LEMMA 6

**Proof**

$$
\begin{aligned}
\|g_{2,m}\| \leq & \left\| \sum_{t=t_m}^{t_{m+1}-1} \alpha_t \left( A^\top A - \mathbb{E}\left[ \hat{A}_{t+f(t)}^\top \hat{A}_t | \mathcal{F}_{t_m+f(t_m)} \right] \right) q_m \right\| \\
& + \left\| \sum_{t=t_m}^{t_{m+1}-1} \alpha_t \mathbb{E}\left[ \hat{A}_{t+f(t)}^\top \left( \hat{b}_t - \hat{A}_t A^{-1} b \right) | \mathcal{F}_{t_m+f(t_m)} \right] \right\| \\
\leq & \sum_{t=t_m}^{t_{m+1}-1} \alpha_t \left\| A^\top A - \mathbb{E}\left[ \hat{A}_{t+f(t)}^\top \hat{A}_t | \mathcal{F}_{t_m+f(t_m)} \right] \right\| \|q_m\| \\
& + \sum_{t=t_m}^{t_{m+1}-1} \alpha_t \left\| \mathbb{E}\left[ \hat{A}_{t+f(t)}^\top \left( \hat{b}_t - \hat{A}_t A^{-1} b \right) | \mathcal{F}_{t_m+f(t_m)} \right] \right\| \\
\leq & \sum_{t=t_m}^{t_{m+1}-1} \alpha_t \left\| A^\top A - \mathbb{E}\left[ \hat{A}_{t+f(t)}^\top \hat{A}_t | \mathcal{F}_{t_m+f(t_m)} \right] \right\| \|q_m\| \\
& + \sum_{t=t_m}^{t_{m+1}-1} \alpha_t \left\| \mathbb{E}\left[ \hat{A}_{t+f(t)}^\top \hat{b}_t - A^\top b | \mathcal{F}_{t_m+f(t_m)} \right] \right\| \\
& + \sum_{t=t_m}^{t_{m+1}-1} \alpha_t \left\| \mathbb{E}\left[ A^\top b - \hat{A}_{t+f(t)}^\top \hat{A}_t A^{-1} b | \mathcal{F}_{t_m+f(t_m)} \right] \right\| \\
= & \sum_{t=t_m}^{t_{m+1}-1} \alpha_t \left\| A^\top A - \mathbb{E}\left[ \hat{A}_{t+f(t)}^\top \hat{A}_t | \mathcal{F}_{t_m+f(t_m)} \right] \right\| \|q_m\| \\
& + \sum_{t=t_m}^{t_{m+1}-1} \alpha_t \left\| \mathbb{E}\left[ \hat{A}_{t+f(t)}^\top \hat{b}_t - A^\top b | \mathcal{F}_{t_m+f(t_m)} \right] \right\| \\
& + \sum_{t=t_m}^{t_{m+1}-1} \alpha_t \left\| \mathbb{E}\left[ A^\top A - \hat{A}_{t+f(t)}^\top \hat{A}_t | \mathcal{F}_{t_m+f(t_m)} \right] A^{-1} b \right\| \\
\leq & \sum_{t=t_m}^{t_{m+1}-1} \alpha_t \left\| A^\top A - \mathbb{E}\left[ \hat{A}_{t+f(t)}^\top \hat{A}_t | \mathcal{F}_{t_m+f(t_m)} \right] \right\| \|q_m\| \\
& + \sum_{t=t_m}^{t_{m+1}-1} \alpha_t \left\| \mathbb{E}\left[ \hat{A}_{t+f(t)}^\top \hat{b}_t - A^\top b | \mathcal{F}_{t_m+f(t_m)} \right] \right\| \\
& + \sum_{t=t_m}^{t_{m+1}-1} \alpha_t \left\| \mathbb{E}\left[ A^\top A - \hat{A}_{t+f(t)}^\top \hat{A}_t | \mathcal{F}_{t_m+f(t_m)} \right] \right\| \left\| A^{-1} b \right\|.
\end{aligned}
$$

To bound the last three terms, we will consider separately whether $t_m + f(t_m) < t_{m+1}$. If $t_m + f(t_m) \geq t_{m+1}$, then applying Lemma 14 yields

$$
\begin{aligned}
\|g_{2,m}\| \leq & \sum_{t=t_m}^{t_{m+1}-1} \alpha_t C_M \|q_m\| + \sum_{t=t_m}^{t_{m+1}-1} \alpha_t C_M + \sum_{t=t_m}^{t_{m+1}-1} \alpha_t C_M H \\
= & C_M (\|q_m\| + H + 1) \sum_{t=t_m}^{t_{m+1}-1} \alpha_t.
\end{aligned}
$$

Since $\alpha_t = \frac{C_\alpha}{(1+t)^\nu}$ is a decreasing function in $t$, we have

$$\|g_{2,m}\| \le C_M(\|q_m\| + H + 1) \sum_{t=t_m}^{t_{m+1}-1} \alpha_{t_m}$$
$$= C_M(\|q_m\| + H + 1)(t_{m+1} - t_m)\alpha_{t_m}$$
$$\le C_M(\|q_m\| + H + 1)f(t_m)\alpha_{t_m}.$$

Assumption 4.4 suggests $f(t) \le C_\tau \alpha_t^{-\tau}$, so

$$\|g_{2,m}\| \le C_M(\|q_m\| + H + 1)C_\tau \alpha_t^{-\tau}\alpha_{t_m}$$
$$\le C_M C_\tau(\|q_m\| + H + 1)\alpha_t^{1-\tau}.$$

Because $\alpha_t \le T_m^2$ for $t \ge t_m$ (Lemma 2),

$$\|g_{2,m}\| \le C_M C_\tau(\|q_m\| + H + 1)T_m^{2(1-\tau)}$$
$$\le C_M C_\tau(H + 1)T_m^{2(1-\tau)}(\|q_m\| + 1).$$

For the general case where $t_m + f(t_m) < t_{m+1}$, we break the summation $\sum_{t=t_m}^{t_{m+1}-1}$ into two parts, i.e. $\sum_{t=t_m}^{t_m+f(t_m)-1}$ and $\sum_{t=t_m+f(t_m)}^{t_{m+1}-1}$ and apply Lemma 14 separately. We have

$$\|g_{2,m}\| \le \sum_{t=t_m}^{t_m+f(t_m)-1} \alpha_t \left\| A^\top A - \mathbb{E}\left[ \hat{A}_{t+f(t)}^\top \hat{A}_t \mid \mathcal{F}_{t_m+f(t_m)} \right] \right\| \|q_m\|$$
$$+ \sum_{t=t_m+f(t_m)}^{t_{m+1}-1} \alpha_t \left\| A^\top A - \mathbb{E}\left[ \hat{A}_{t+f(t)}^\top \hat{A}_t \mid \mathcal{F}_{t_m+f(t_m)} \right] \right\| \|q_m\|$$
$$+ \sum_{t=t_m}^{t_m+f(t_m)-1} \alpha_t \left\| \mathbb{E}\left[ \hat{A}_{t+f(t)}^\top \hat{b}_t - A^\top b \mid \mathcal{F}_{t_m+f(t_m)} \right] \right\|$$
$$+ \sum_{t=t_m+f(t_m)}^{t_{m+1}-1} \alpha_t \left\| \mathbb{E}\left[ \hat{A}_{t+f(t)}^\top \hat{b}_t - A^\top b \mid \mathcal{F}_{t_m+f(t_m)} \right] \right\|$$
$$+ \sum_{t=t_m}^{t_m+f(t_m)-1} \alpha_t \left\| \mathbb{E}\left[ A^\top A - \hat{A}_{t+f(t)}^\top \hat{A}_t \mid \mathcal{F}_{t_m+f(t_m)} \right] \right\| \|A^{-1}b\|$$
$$+ \sum_{t=t_m+f(t_m)}^{t_{m+1}-1} \alpha_t \left\| \mathbb{E}\left[ A^\top A - \hat{A}_{t+f(t)}^\top \hat{A}_t \mid \mathcal{F}_{t_m+f(t_m)} \right] \right\| \|A^{-1}b\|$$
$$\le \sum_{t=t_m}^{t_m+f(t_m)-1} \alpha_t C_M \|q_m\| + \sum_{t=t_m+f(t_m)}^{t_{m+1}-1} \alpha_t C_M \left( \chi^{f(t)} + \chi^{t-[t_m+f(t_m)]} \right) \|q_m\|$$
$$+ \sum_{t=t_m}^{t_m+f(t_m)-1} \alpha_t C_M + \sum_{t=t_m+f(t_m)}^{t_{m+1}-1} \alpha_t C_M \left( \chi^{f(t)} + \chi^{t-[t_m+f(t_m)]} \right)$$
$$+ \sum_{t=t_m}^{t_m+f(t_m)-1} \alpha_t C_M H + \sum_{t=t_m+f(t_m)}^{t_{m+1}-1} \alpha_t C_M \left( \chi^{f(t)} + \chi^{t-[t_m+f(t_m)]} \right) H$$
$$\le C_M(\|q_m\| + H + 1)\left[ \sum_{t=t_m}^{t_m+f(t_m)-1} \alpha_t + \sum_{t=t_m+f(t_m)}^{t_{m+1}-1} \alpha_t \left( \chi^{f(t)} + \chi^{t-[t_m+f(t_m)]} \right) \right].$$

Because $\alpha_t = \frac{C_\alpha}{(1+t)^\nu}$ is a decreasing function in $t$ and $\alpha_t \leq T_m^2$ for $t \geq t_m$ (Lemma 2), we get

$$
\begin{aligned}
\|g_{2,m}\| &\leq C_M(\|q_m\| + H + 1) \left[ \sum_{t=t_m}^{t_m+f(t_m)-1} \alpha_{t_m} + T_m^2 \sum_{t=t_m+f(t_m)}^{t_{m+1}-1} \left( \chi^{f(t)} + \chi^{t-[t_m+f(t_m)]} \right) \right] \\
&\leq C_M(\|q_m\| + H + 1) \left[ f(t_m)\alpha_{t_m} + T_m^2 \left( \sum_{t=t_m+f(t_m)}^{t_{m+1}-1} \chi^{f(t)} + \sum_{t=t_m+f(t_m)}^{t_{m+1}-1} \chi^{t-[t_m+f(t_m)]} \right) \right] \\
&\leq C_M(\|q_m\| + H + 1) \left[ f(t_m)\alpha_{t_m} + T_m^2 \left( \sum_{t=0}^{\infty} \chi^{f(t)} + \sum_{t=0}^{\infty} \chi^t \right) \right]
\end{aligned}
$$

Assumption 4.4 suggests that $f(t) \leq C_\tau \alpha_t^{-\tau}$ and $L(f,\chi) = \sum_{t=0}^{\infty} \chi^{f(t)} < \infty$. Therefore,

$$
\begin{aligned}
\|g_{2,m}\| &\leq C_M(\|q_m\| + H + 1) \left[ C_\tau \alpha_{t_m}^{-\tau} \alpha_{t_m} + T_m^2 \left( L(f,\chi) + \sum_{t=0}^{\infty} \chi^t \right) \right] \\
&\leq C_M(\|q_m\| + H + 1) \left[ C_\tau \alpha_{t_m}^{1-\tau} + T_m^2 \left( L(f,\chi) + \frac{1}{1-\chi} \right) \right] \\
&\leq C_M(\|q_m\| + H + 1) \left[ C_\tau T_m^{2(1-\tau)} + T_m^2 \left( L(f,\chi) + \frac{1}{1-\chi} \right) \right].
\end{aligned}
$$

When $T_m \leq 1$, we then have

$$
\|g_{2,m}\| \leq C_M T_m^{2(1-\tau)}(H+1) \left( C_\tau + L(f,\chi) + \frac{1}{1-\chi} \right) (\|q_m\| + 1),
$$

which completes the proof. ∎

### C.6 PROOF OF LEMMA 7

**Proof** Firstly, we have

$$
\|g_{3,m}\| \leq \left\| \sum_{t=t_m}^{t_{m+1}-1} \alpha_t \left( \mathbb{E}\left[ \hat{A}_{t+f(t)}^\top \hat{A}_t | \mathcal{F}_{t_m+f(t_m)} \right] - \hat{A}_{t+f(t)}^\top \hat{A}_t \right) q_m \right\|
$$
$$
+ \left\| \sum_{t=t_m}^{t_{m+1}-1} \alpha_t \left( \mathbb{E}\left[ \hat{A}_{t+f(t)}^\top \left( \hat{b}_t - \hat{A}_t A^{-1} b \right) | \mathcal{F}_{t_m+f(t_m)} \right] - \hat{A}_{t+f(t)}^\top \left( \hat{b}_t - \hat{A}_t A^{-1} b \right) \right) \right\|
$$

$$
\leq \sum_{t=t_m}^{t_{m+1}-1} \alpha_t \left( \left\| \mathbb{E}\left[ \hat{A}_{t+f(t)}^\top \hat{A}_t | \mathcal{F}_{t_m+f(t_m)} \right] \right\| + \left\| \hat{A}_{t+f(t)}^\top \right\| \left\| \hat{A}_t \right\| \right) \|q_m\|
$$
$$
+ \sum_{t=t_m}^{t_{m+1}-1} \alpha_t \left( \left\| \mathbb{E}\left[ \hat{A}_{t+f(t)}^\top \left( \hat{b}_t - \hat{A}_t A^{-1} b \right) | \mathcal{F}_{t_m+f(t_m)} \right] \right\| + \left\| \hat{A}_{t+f(t)}^\top \right\| \left( \|\hat{b}_t\| + \left\| \hat{A}_t A^{-1} b \right\| \right) \right)
$$

$$
\leq \sum_{t=t_m}^{t_{m+1}-1} \alpha_t \left( \mathbb{E}\left[ \left\| \hat{A}_{t+f(t)}^\top \hat{A}_t \right\| | \mathcal{F}_{t_m+f(t_m)} \right] + H \cdot H \right) \|q_m\|
$$
$$
+ \sum_{t=t_m}^{t_{m+1}-1} \alpha_t \left( \mathbb{E}\left[ \left\| \hat{A}_{t+f(t)}^\top \left( \hat{b}_t - \hat{A}_t A^{-1} b \right) \right\| | \mathcal{F}_{t_m+f(t_m)} \right] + H(H + H) \right)
$$
$$
\leq \sum_{t=t_m}^{t_{m+1}-1} \alpha_t \left( \mathbb{E}\left[ \left\| \hat{A}_{t+f(t)}^\top \right\| \left\| \hat{A}_t \right\| | \mathcal{F}_{t_m+f(t_m)} \right] + H^2 \right) \|q_m\|
$$
$$
+ \sum_{t=t_m}^{t_{m+1}-1} \alpha_t \left( \mathbb{E}\left[ \left\| \hat{A}_{t+f(t)}^\top \right\| \left( \|\hat{b}_t\| + \left\| \hat{A}_t A^{-1} b \right\| \right) | \mathcal{F}_{t_m+f(t_m)} \right] + 2H^2 \right)
$$
$$
\leq \sum_{t=t_m}^{t_{m+1}-1} \alpha_t \left( H^2 + H^2 \right) \|q_m\| + \sum_{t=t_m}^{t_{m+1}-1} \alpha_t \left( H(H + H) + 2H^2 \right)
$$
$$
\leq 2\bar{\alpha}_m H^2 (\|q_m\| + 2)
$$
$$
\leq 4T_m H^2 (\|q_m\| + 2) \qquad\qquad\qquad\qquad \text{(Lemma 3)}
$$
$$
\leq 8T_m H^2 (\|q_m\| + 1).
$$

Secondly,

$$
\mathbb{E}\left[ g_{3,m} | \mathcal{F}_{t_m+f(t_m)} \right] = 0
$$

holds trivially, which completes the proof. ∎

### C.7 PROOF OF LEMMA 8

**Proof**

$$
\|g_{4,m}\| = \left\| \sum_{t=t_m}^{t_{m+1}-1} \alpha_t \hat{A}_{t+f(t)}^\top \hat{A}_t \left[ q_m - (w_t + A^{-1} b) \right] \right\|
$$
$$
\leq \sum_{t=t_m}^{t_{m+1}-1} \alpha_t \left\| \hat{A}_{t+f(t)}^\top \right\| \|\hat{A}_t\| \|q_m - (w_t + A^{-1} b)\|.
$$

Then, by Lemma 4, we have

$$\|g_{4,m}\| \leq \sum_{t=t_m}^{t_{m+1}-1} \alpha_t H \cdot H \cdot 8 T_m H^2 (\|q_m\| + 1)$$
$$= 8 \bar{\alpha}_m T_m H^4 (\|q_m\| + 1),$$

which completes the proof. ∎

## C.8 PROOF OF LEMMA 9

**Proof**

$$
\begin{aligned}
\|q_{m+1}\|^2 =& \|q_m + g_{1,m} + g_{2,m} + g_{3,m} + g_{4,m}\|^2 \\
=& \|q_m + g_{1,m}\|^2 + \|g_{2,m} + g_{3,m} + g_{4,m}\|^2 + 2(q_m + g_{1,m})^\top (g_{2,m} + g_{3,m} + g_{4,m}) \\
\leq& \|q_m + g_{1,m}\|^2 + \|g_{2,m} + g_{3,m} + g_{4,m}\|^2 \\
& + 2(q_m + g_{1,m})^\top g_{3,m} + 2\|q_m + g_{1,m}\|\|g_{2,m} + g_{4,m}\|.
\end{aligned}
$$

Lemma 5 implies that $\|q_m + g_{1,m}\|^2 \leq (1 - \beta T_m)\|q_m\|^2$ and $\|q_m + g_{1,m}\| = \mathcal{O}(\|q_m\|)$, Lemma 6 suggests that $\|g_{2,m}\| = \mathcal{O}\left(T_m^{2(1-\tau)}(\|q_m\| + 1)\right)$, Lemma 7 suggests that $\|g_{3,m}\| = \mathcal{O}(T_m(\|q_m\| + 1))$, and Lemma 8 suggests that $\|g_{4,m}\| = \mathcal{O}\left(T_m^2(\|q_m\| + 1)\right)$. Hence,

$$\|g_{2,m} + g_{3,m} + g_{4,m}\| \leq \|g_{2,m}\| + \|g_{3,m}\| + \|g_{4,m}\| = \mathcal{O}(T_m(\|q_m\| + 1)),$$

and

$$\|g_{2,m} + g_{4,m}\| = \mathcal{O}\left(T_m^{2(1-\tau)}(\|q_m\| + 1)\right).$$

Moreover, both $q_m$ and $g_{1,m} = -\bar{\alpha}_m A^\top A q_m$ are adapted to $\mathcal{F}_{t_m + f(t_m)}$ and 7 implies that $\mathbb{E}\left[g_{3,m}|\mathcal{F}_{t_m + f(t_m)}\right] = 0$. We, therefore, have

$$\mathbb{E}\left[(q_m + g_{1,m})^\top g_{3,m}|\mathcal{F}_{t_m + f(t_m)}\right] = 0.$$

Lastly, putting everything together, we have

$$
\begin{aligned}
& \mathbb{E}\left[\|q_{m+1}\|^2 | \mathcal{F}_{t_m + f(t_m)}\right] \\
\leq & (1 - \beta T_m)\|q_m\|^2 + \mathcal{O}(T_m(\|q_m\| + 1))^2 + \mathcal{O}(\|q_m\|)\mathcal{O}\left(T_m^{2(1-\tau)}(\|q_m\| + 1)\right) \\
\leq & (1 - \beta T_m)\|q_m\|^2 + \mathcal{O}\left(T_m^{2(1-\tau)}(\|q_m\| + 1)^2\right) \\
\leq & (1 - \beta T_m)\|q_m\|^2 + \mathcal{O}\left(T_m^{2(1-\tau)}(\|q_m\|^2 + 1)\right),
\end{aligned}
$$

where the last inequality comes from the fact that $(\|q_m\| + 1)^2 \leq 2(\|q_m\|^2 + 1)$. In conclusion, there exists a constant $D$ such that

$$\mathbb{E}\left[\|q_{m+1}\|^2 | \mathcal{F}_{t_m + f(t_m)}\right] \leq \left(1 - \beta T_m + D T_m^{2(1-\tau)}\right)\|q_m\|^2 + D T_m^{2(1-\tau)},$$

which completes the proof. ∎

## C.9 PROOF OF LEMMA 10

**Proof** To prove the lemma, we will invoke a supermartingale convergence theorem stated as follows.

**Theorem 16** *(Proposition 4.2 in Bertsekas & Tsitsiklis (1996)) Let $Y_m$, $X_m$, and $Z_m$, $m \geq 0$ be three sequences of random variables and let $\bar{\mathcal{F}}_m$, $m \geq 0$, be sets of random variables such that $\bar{\mathcal{F}}_m \subseteq \bar{\mathcal{F}}_{m+1}$ for all $m$. Suppose that*

1. The random variables $Y_m$, $X_m$, and $Z_m$ are non-negative and are functions of the random variables in $\bar{\mathcal{F}}_m$,

2. For each $m$, we have $\mathbb{E}\left[Y_{m+1}|\bar{\mathcal{F}}_m\right] \leq Y_m - X_m + Z_m$,

3. There holds $\sum_{m=0}^{\infty} Z_m < \infty$.

Then, we have $\sum_{m=0}^{\infty} X_m < \infty$ almost surely, and the sequence $Y_m$ converges almost surely to a non-negative random variable $Y$.

In our case, we let

$$
\begin{aligned}
Y_m =& \|q_m\|^2, \\
X_m =& \frac{1}{2}\beta T_m \|q_m\|^2, \\
Z_m =& DT_m^{2(1-\tau)}, \\
\bar{\mathcal{F}}_m =& \mathcal{F}_{t_m+f(t_m)}.
\end{aligned}
$$

The first condition of Theorem 16 holds trivially. For the second condition of Theorem 16 to hold, we rely on (18) in Lemma 9. According to the definition of $T_m$ in (12), we have

$$
\lim_{m\to\infty} T_m = 0.
$$

As a result, the condition

$$
T_m \leq \min\left(\frac{\beta}{2H^4}, 1, \frac{\ln(2)}{2H^2}\right)
$$

in Lemma 9 holds for sufficiently large $m$. Moreover, since

$$
\begin{aligned}
\tau <& \frac{3}{2} - \frac{1}{\nu} && \text{(Assumption 4.4)} \\
\leq& \frac{3}{2} - \frac{1}{1} && \text{(Assumption 4.3)} \\
=& \frac{1}{2},
\end{aligned}
$$

we have

$$
2(1-\tau) > 1.
$$

Consequently, the condition

$$
DT_m^{2(1-\tau)} \leq \frac{1}{2}\beta T_m,
$$

which is equivalent to

$$
T_m^{2(1-\tau)-1} \leq \frac{\beta}{2D},
$$

also holds for sufficiently large $m$ as $\lim_{m\to\infty} T_m = 0$. Crucially, $D$ is deterministic because $D$ only depends on $H$, $\beta$, $C_\tau$, $C_M$, $L(f,\chi)$, and $\chi$, which are all deterministic quantities. Therefore, there always exists a finite and deterministic $m_0$ such that the subsequence $\{X_m, Y_m, Z_m\}_{m\geq m_0}$ verifies the second condition. Since $\eta > \frac{1}{2(1-\tau)}$, i.e. $2(1-\tau)\eta > 1$, by p-test, we can deduce that

$$
\begin{aligned}
\sum_{m=0}^{\infty} Z_m =& \sum_{m=0}^{\infty} DT_m^{2(1-\tau)} = D\sum_{m=0}^{\infty}\left(\frac{16\max(C_\alpha,1)}{(\eta+1)(m+1)^\eta}\right)^{2(1-\tau)} \\
=& \frac{(16\max(C_\alpha,1))^{2(1-\tau)}D}{(\eta+1)^{2(1-\tau)}}\sum_{m=0}^{\infty}\frac{1}{(\eta+1)^{2(1-\tau)\eta}} < \infty.
\end{aligned}
$$

The third condition of Theorem 16, therefore, also holds. All the conditions of Theorem 16 are now verified for the subsequence $\{X_m, Y_m, Z_m\}_{m \geq m_0}$, which implies that the sequence $\left\{\|q_m\|^2\right\}_{m \geq 0}$ converges and

$$\frac{1}{2}\beta \sum_{m=0}^{\infty} T_m \|q_m\|^2 < \infty.$$

As $\eta < 1$, we have

$$\sum_{m=0}^{\infty} T_m = \sum_{m=0}^{\infty} \frac{16 \max(C_\alpha, 1)}{(\eta+1)(m+1)^\eta} = \frac{16 \max(C_\alpha, 1)}{\eta+1} \sum_{m=0}^{\infty} \frac{1}{(m+1)^\eta} = \infty.$$

Thus, $\|q_m\|^2$ must converge to zero. Otherwise, $\sum_{m=0}^{\infty} X_m = \frac{1}{2}\beta \sum_{m=0}^{\infty} T_m \|q_m\|^2$ diverges to infinity. Thus, $q_m$ converges to 0 almost surely, which completes the proof. ∎

## C.10 PROOF OF LEMMA 12

**Proof** First, we prove the bounds for each function.

**Bound for $g_t(\cdot)$:**

$$\|g_t(w)\| = \left\|-\hat{A}_{t+f(t)}^\top(\hat{A}_t w + \hat{b}_t)\right\| \leq \left\|\hat{A}_{t+f(t)}^\top\right\|(\|\hat{A}_t\|\|w\| + \|\hat{b}_t\|)$$
$$\leq H(HB + H) = H^2(B+1).$$

**Bound for $\bar{g}(\cdot)$:**

$$\|\bar{g}(w)\| = \left\|A^\top(Aw + b)\right\| \leq \left\|A^\top\right\|(\|A\|\|w\| + \|b\|) \leq H(HB + B) = H^2(B+1).$$

**Bound for $\Lambda_t(\cdot)$:**

$$|\Lambda_t(w)| = |\langle w - w_*, g_t(w) - \bar{g}(w)\rangle| \leq \|w - w_*\|\|g_t(w) - \bar{g}(w)\|$$
$$\leq (\|w\| + \|w_*\|)(\|g_t(w)\| - \|\bar{g}(w)\|) \leq (B + B)[H^2(B+1) + H^2(B+1)]$$
$$= 4H^2 B(B+1).$$

Hence, by taking $C_g = \max\{H^2(B+1), 4H^2 B(B+1)\}$, we have the result stated. Second, we prove the functions are Lipschitz.

$g_t(\cdot)$ **is $H^2$-Lipschitz:**

$$\|g_t(w) - g_t(w')\| = \left\|-\hat{A}_{t+f(t)}^\top(\hat{A}_t w + \hat{b}_t) + \hat{A}_{t+f(t)}^\top(\hat{A}_t w' + \hat{b}_t)\right\| = \left\|\hat{A}_{t+f(t)}^\top \hat{A}_t(w - w')\right\|$$
$$\leq \left\|\hat{A}_{t+f(t)}^\top\right\|\left\|\hat{A}_t\right\|\|w - w'\| \leq H^2\|w - w'\|$$

$\bar{g}(\cdot)$ **is $H^2$-Lipschitz:**

$$\|\bar{g}(w) - \bar{g}(w')\| = \left\|-A^\top(Aw + b) + A^\top(Aw' + b)\right\| = \left\|A^\top A(w' - w)\right\|$$
$$\leq \left\|A^\top\right\|\|A\|\|w' - w\| \leq H^2\|w' - w\|.$$

$\Lambda_t(\cdot)$ **is** $2H^2(3B+1)$-**Lipschitz**:

$$
\begin{aligned}
&|\Lambda_t(w) - \Lambda_t(w')| \\
=&|\langle w - w_*, g_t(w) - \bar{g}(w)\rangle - \langle w' - w_*, g_t(w') - \bar{g}(w')\rangle| \\
=&|\langle w - w_*, g_t(w) - \bar{g}(w) - (g_t(w') - \bar{g}(w'))\rangle + \langle w - w_* - (w' - w_*), g_t(w') - \bar{g}(w')\rangle| \\
\leq&|\langle w - w_*, g_t(w) - \bar{g}(w) - (g_t(w') - \bar{g}(w'))\rangle| + |\langle w - w', g_t(w') - \bar{g}(w')\rangle| \\
\leq&\|w - w_*\|\|g_t(w) - \bar{g}(w) - (g_t(w') - \bar{g}(w'))\| + \|w - w'\|\|g_t(w') - \bar{g}(w')\| \\
\leq&(\|w\| + \|w_*\|)(\|g_t(w') - g_t(w)\| + \|\bar{g}(w) - \bar{g}(w')\|) + (\|g_t(w')\| + \|\bar{g}(w')\|)\|w - w'\| \\
\leq&(B + B)(\|g_t(w') - g_t(w)\| + \|\bar{g}(w) - \bar{g}(w')\|) + [H^2(B+1) + H^2(B+1)]\|w - w'\| \\
\leq&2B(\|g_t(w') - g_t(w)\| + \|\bar{g}(w) - \bar{g}(w')\|) + 2H^2(B+1)\|w - w'\| \\
\leq&2B(H^2\|w' - w\| + H^2\|w' - w\|) + 2H^2(B+1)\|w - w'\| \\
=&2H^2(3B+1)\|w - w'\|.
\end{aligned}
$$

Therefore, by taking $C_{Lip} = \max\{H^2, 2H^2(3B+1)\}$, we have the result stated.

Lastly, we prove the following inequality regarding the inner product.

$$
\begin{aligned}
\langle w - w', \bar{g}(w) - \bar{g}(w')\rangle &= \langle w - w', -A^\top(Aw + b) + A^\top(Aw' + b)\rangle \\
&= -(w - w')^\top A^\top A(w - w') \leq -\beta\|w - w'\|^2,
\end{aligned}
$$

where the last inequality holds due to (17). $\blacksquare$

### C.11 PROOF OF LEMMA 13

**Proof** For any $i \geq 0$, since $w_i$ lies in the ball for projection, we have

$$
\begin{aligned}
\|w_{i+1} - w_i\| = \|\Gamma(w_i + \alpha_i g_i(w_i)) - w_i\| \\
= \|\Gamma(w_i + \alpha_i g_i(w_i)) - \Gamma(w_i)\| \\
\leq \|\Gamma(w_i + \alpha_i g_i(w_i) - w_i)\| \qquad (\Gamma(\cdot) \text{ is non-expansive}) \\
= \|\Gamma(\alpha_i g_i(w_i))\| \\
\leq \|\alpha_i g_i(w_i)\| \\
= \alpha_i\|g_i(w_i)\| \\
\leq \alpha_i C_g. \qquad (\text{Lemma } 12)
\end{aligned}
$$

Therefore, by telescoping, we can deduce that for all $0 < k < t$,

$$
\|w_t - w_{t-k}\| \leq \sum_{i=t-k}^{t-1} \|w_{i+1} - w_i\| \leq \sum_{i=t-k}^{t-1} C_g\alpha_i = C_g\sum_{i=t-k}^{t-1}\alpha_i.
$$

Because $\frac{1}{t+1}$ is a decreasing function in $t$, for all $i \in [t, t+1]$, $\frac{1}{t+1} \leq \frac{1}{i}$. As a consequence, $\frac{1}{t+1} \leq \int_t^{t+1}\frac{1}{i}$, and

$$
\sum_{i=t-k}^{t}\frac{1}{i+1} \leq \sum_{i=t-k}^{t}\int_i^{i+1}\frac{1}{j} \leq \int_{t-k}^{t}\frac{1}{i} = \ln(t) - \ln(t-k) = \ln\left(\frac{t}{t-k}\right).
$$

In addition, given that $\alpha_t = \frac{C_\alpha}{t+1}$, we have

$$
\|w_t - w_{t-k}\| \leq C_g\sum_{i=t-k}^{t-1}\alpha_i = C_g\sum_{i=t-k}^{t-1}\frac{C_\alpha}{i+1} \leq C_gC_\alpha\ln\left(\frac{t}{t-k}\right).
$$

Applying Lemma 12 again, we get

$$
|\Lambda_t(w_t) - \Lambda_t(w_{t-k})| \leq C_gC_\alpha C_{Lip}\ln\left(\frac{t}{t-k}\right)
$$

and hence

$$\Lambda_t(w_t) \leq \Lambda_t(w_{t-k}) + C_g C_\alpha C_{Lip} \ln\left(\frac{t}{t-k}\right).$$

(29)

We now bound the expectation of $\Lambda_t(w_{t-k})$ conditioning on $\mathcal{F}_{t-k+f(t-k)}$. In particular, we have

$$\begin{aligned}
&\mathbb{E}\left[\Lambda_t(w_{t-k})|\mathcal{F}_{t-k+f(t-k)}\right] \\
=&\mathbb{E}\left[\langle w_{t-k} - w_*, g_t(w_{t-k}) - \bar{g}(w_{t-k})\rangle|\mathcal{F}_{t-k+f(t-k)}\right] \\
=&\mathbb{E}\left[\left\langle w_{t-k} - w_*, \hat{A}_{t+f(t)}^\top(\hat{A}_t w_{t-k} + \hat{b}_t) - A^\top(Aw_{t-k} - b)\right\rangle|\mathcal{F}_{t-k+f(t-k)}\right] \\
=&\mathbb{E}\left[\left\langle w_{t-k} - w_*, \left(\hat{A}_{t+f(t)}^\top\hat{A}_t - A^\top A\right) w_{t-k} - \left(\hat{A}_{t+f(t)}^\top\hat{b}_t - A^\top b\right)\right\rangle|\mathcal{F}_{t-k+f(t-k)}\right].
\end{aligned}$$

As expectation and dot product are linear and $w_{t-k}$ is adapted to $\mathcal{F}_{t-k+f(t-k)}$, we can further reduce our expectations as

$$\begin{aligned}
&\mathbb{E}\left[\Lambda_t(w_{t-k})|\mathcal{F}_{t-k+f(t-k)}\right] \\
=&\left\langle w_{t-k} - w_*, \left(\mathbb{E}\left[\hat{A}_{t+f(t)}^\top\hat{A}_t|\mathcal{F}_{t-k+f(t-k)}\right] - A^\top A\right) w_{t-k} - \left(\mathbb{E}\left[\hat{A}_{t+f(t)}^\top\hat{b}_t|\mathcal{F}_{t-k+f(t-k)}\right] - A^\top b\right)\right\rangle \\
\leq&\|w_{t-k} - w_*\|\left\|\left(\mathbb{E}\left[\hat{A}_{t+f(t)}^\top\hat{A}_t|\mathcal{F}_{t-k+f(t-k)}\right] - A^\top A\right) w_{t-k} - \left(\mathbb{E}\left[\hat{A}_{t+f(t)}^\top\hat{b}_t|\mathcal{F}_{t-k+f(t-k)}\right] - A^\top b\right)\right\| \\
\leq&(\|w_{t-k}\| + \|w_*\|)\left(\left\|\left(\mathbb{E}\left[\hat{A}_{t+f(t)}^\top\hat{A}_t|\mathcal{F}_{t-k+f(t-k)}\right] - A^\top A\right) w_{t-k}\right\| + \left\|\mathbb{E}\left[\hat{A}_{t+f(t)}^\top\hat{b}_t|\mathcal{F}_{t-k+f(t-k)}\right] - A^\top b\right\|\right) \\
\leq&(\|w_{t-k}\| + \|w_*\|)\left(\left\|\mathbb{E}\left[\hat{A}_{t+f(t)}^\top\hat{A}_t|\mathcal{F}_{t-k+f(t-k)}\right] - A^\top A\right\|\|w_{t-k}\| + \left\|\mathbb{E}\left[\hat{A}_{t+f(t)}^\top\hat{b}_t|\mathcal{F}_{t-k+f(t-k)}\right] - A^\top b\right\|\right) \\
\leq&(B + B)\left(\left\|\mathbb{E}\left[\hat{A}_{t+f(t)}^\top\hat{A}_t|\mathcal{F}_{t-k+f(t-k)}\right] - A^\top A\right\|B + \left\|\mathbb{E}\left[\hat{A}_{t+f(t)}^\top\hat{b}_t|\mathcal{F}_{t-k+f(t-k)}\right] - A^\top b\right\|\right) \\
\leq&2B\left(\left\|\mathbb{E}\left[\hat{A}_{t+f(t)}^\top\hat{A}_t|\mathcal{F}_{t-k+f(t-k)}\right] - A^\top A\right\|B + \left\|\mathbb{E}\left[\hat{A}_{t+f(t)}^\top\hat{b}_t|\mathcal{F}_{t-k+f(t-k)}\right] - A^\top b\right\|\right).
\end{aligned}$$

Applying Lemma 14, we get

$$\mathbb{E}[\Lambda_t(w_{t-k})|\mathcal{F}_{t-k+f(t-k)}] \leq \quad 2B(B+1)C_M \begin{cases} 1 & t < t-k+f(t-k) \\ \chi^{f(t)} + \chi^{t-(t-k+f(t-k))} & t-k+f(t-k) \leq t \end{cases}.$$

Taking total expectations then yields

$$\mathbb{E}[\Lambda_t(w_{t-k})] \leq 2B(B+1)C_M \begin{cases} 1 & k < f(t-k) \\ \chi^{f(t)} + \chi^{t-(t-k+f(t-k))} & k \geq f(t-k) \end{cases}.$$

(30)

Plugging (30) into the expectation of (29) yields

$$\mathbb{E}[\Lambda_t(w_t)] \leq \quad C_g C_\alpha C_{Lip} \ln\left(\frac{t}{t-k}\right) + 2B(B+1)C_M \begin{cases} 1 & k < f(t-k) \\ \chi^{f(t)} + \chi^{t-(t-k+f(t-k))} & k \geq f(t-k) \end{cases},$$

which completes the proof. ∎

