# OpenReview forum: "Revisiting a Design Choice in Gradient Temporal Difference Learning"
_ICLR.cc/2025/Conference — ICLR 2025 Poster_

### Official Review · Reviewer_R5Yw · 2024-10-26

**Soundness:** 3
**Presentation:** 3
**Contribution:** 3
**Rating:** 6
**Confidence:** 3

**Summary:**

This paper proposed a new solution to solve double sampling issue in off-policy reinforcement learning. Specifically, this paper provided another method to estimate $A^T A$ and $A^T b$ by introducing a function $f(t)$. The authors also provided finite-time analysis for their method as well as some numerical results.

**Strengths:**

The paper expand the idea from $A^TTD$ and proposed a new method to solve double sampling issue in off-policy learning. Compared with $A^TTD$, this methods required less memory. The authors also provided the convergence analysis of their method.

**Weaknesses:**

This paper is really interesting to me. However, I have several questions.

1. The advantage of selecting $f(t)$ as an increasing function over a constant one is not immediately clear. The authors state in lines 186–187 that classical convergence analysis can be applied to establish the convergence rate. Thus, it seems that a constant $f(t)$ could also ensure convergence. Additionally, the experimental results suggest that setting $f(t)=2$ is sufficient to resolve Baird’s counterexample, which further supports the idea of choosing $f(t)$ as a constant.

2. Relying on samples from several steps prior may introduce additional errors during policy improvement. Although this paper focuses exclusively on policy evaluation, it is important to mark that policy evaluation serves the purpose of policy improvement. In cases where the policy is continuously updated, the samples used to estimate $A^T$ may become inaccurate, introducing further errors.

3. A minor issue appears in lines 206–207, where I believe the correct notation should be $\bar{h}(\omega_{t_m})$ instead of $h(\omega_{t_m})$

**Questions:**

Please see my comments in Weaknesses section.

---

> ### Author Response · Authors · 2024-11-25
>
> > Thus, it seems that a constant $f(t)$ could also ensure convergence.
>
> No it won't. If it's constant, we would have $E[\hat A_{t+t_0}^\top \hat A_t]$. But this matrix is not positive definite because it's not $A^\top A$ due to the correlation. So the analysis will not go through. We added a footnote in page 4 to further clarify this.
>
> >  Additionally, the experimental results suggest that setting $f(t) = 2$ is sufficient to resolve Baird’s counterexample, which further supports the idea of choosing $f(t)$ as a constant.
>
> It is true that a constant $f(t)$ works in **some** tasks. But to ensure it works for **all** tasks in the worst case, we have to use an increasing $f(t)$.
>
> > it is important to mark that policy evaluation serves the purpose of policy improvement
>
> We fully agree with the reviewer on this point. One easy solution is that we only update policy every $f(t)$ steps. Since $f(t)$ is really small, we expect that this will not affect the performance much.
>
> > A minor issue appears in lines 206–207
>
> Thanks for pointing this out. We have fixed this.

---

> > ### Comment · Reviewer_R5Yw · 2024-12-01
> > **Respond to authors**
> >
> > Thanks for the authors' reply.
> >
> > However, I am still not clear on why the gap function $f(t)$ must be increasing. While I agree with your argument that $E[\hat A_{t+t_0}^\top \hat A_t]$ is not positive definite, it seems that $E [\hat{A}_{t+f(t)}^T \hat{A}_t]$ is also not positive definite. This does not fully address the question of why your method works with a slowly increasing $f(t)$ instead of a constant.

---

> ### Author Response · Authors · 2024-12-02
>
> Sorry for the confusion. We should have clarified more that despite $E[\hat A^\top_{t+f(t)} \hat A_t]$ is not positive definite, this expectation converges to $A^\top A$ as $t \to \infty$, which is positive definite. In other words, it is true that $\hat A^\top_{t+f(t)} \hat A_t$ is not an unbiased estimator for $A^\top A$ for any finite $t$, but it is consistent. By contrast, $\hat A^\top_{t+t_0} \hat A_t$ is always an biased estimator for any finite $t_0$ and is not consistent because $t_0$ is finite.

---

> > ### Comment · Reviewer_R5Yw · 2024-12-03
> > **Respond to authors**
> >
> > Thank you for the clarification. I now have a much clearer understanding of the ideas in this paper.
> >
> > I also share Reviewer N6bs's observation that $f(t)$ appears to have a relationship with the mixing time (Bhandari et al. (2018)). Intuitively, if $f(t)$ exceeds the mixing time, $s_{t+f(t)}$ becomes nearly independent of $s_t$ and therefore the bias of the estimation $E[\hat A^\top_{t+f(t)} \hat A_t]$ will be significantly reduced. While the authors argued in their response to Reviewer N6bs that setting $f(t)$ based on the mixing time is not practical, I believe this connection would be worthwhile to explore from a theoretical perspective.
> >
> > At this point, I am inclined to maintain my score. However, I would consider increasing my score if the authors further investigated the relationship between $f(t)$ and the mixing time.
> >
> > Reference:
> >
> > Bhandari, J., Russo, D., and Singal, R. A finite time analysis of temporal difference learning with linear function approximation. In Conference On Learning Theory, pp. 1691–1692, 2018.

---

> > > ### Author Response · Authors · 2024-12-03
> > >
> > > Thanks for the response.
> > >
> > > >  if the authors further investigated the relationship between $f(t)$ and the mixing time.
> > >
> > > We believe the Eq (24) in Lemma 14 in page 16 might be what the reviewer is looking for. (We set $k=f(t)$ when invoking that lemma). Eq (24) precisely quantifies the bias resulting from using $f(t)$, where $\chi$ is the mixing factor of the chain. As can be seen in Eq (24), this bias diminishes geometrically for whatever $\chi$. So eventually, when combining that bias with other polynomial error terms, that bias gets dominated by the polynomial terms and are hidden by the big O notation.

---

> ### Author Response · Authors · 2024-12-03
>
> Or more precisely speaking, the last time that the relationship between $f(t)$ and the mixing time appears is in Lemma 6, the $L(f, \chi)$ term. But it is a constant so gets hidden in the following analysis. The $f(t)$ indeed needs to be set according to the mixing factor $\chi$ such that $\sum \chi^{f(t)} < \infty$. But as long as we know the chain is geometrically mixing, we can set $f(t)$ without knowing the exact value of $\chi$. That being said, if we do know the mixing factor $\chi$, we can optimize the choice of $f(t)$ accordingly to use the minimal possible $f(t)$ such that $\sum \chi^{f(t)}$ remains finite.

---

> > ### Comment · Reviewer_R5Yw · 2024-12-03
> > **Respond to authors**
> >
> > Thank you for pointing that out. Indeed, Eq. (24) in Lemma 14 on page 16 has clarified this issue for me, and I no longer have further questions. As mentioned earlier, I will increase my score to 6 at this moment.

---

### Official Review · Reviewer_N6bs · 2024-10-31

**Soundness:** 3
**Presentation:** 2
**Contribution:** 2
**Rating:** 8
**Confidence:** 3

**Summary:**

This paper proposes a new variant of the gradient temporal difference (GTD) learning algorithm for online off-policy policy evaluation with linear function approximation. The idea is to use two samples distanced away from each other to address the double sampling issue encountered during the derivation of GTD. The paper shows that when the distance between the two samples $f(t)$ used to estimate the gradient increases with a proper rate (e.g., $f(t)=\ln^2(t+1)$), the new algorithm converges asymptotically, while its variant with a projection operator has a convergent rate comparable to on-policy TD. The consequence of this new GTD variant is that 1) it reduces the need for an additional set of weights and step size, and 2) it requires an additional memory of size $O(\ln^2(t))$. Preliminary experiment results on Baird’s counterexample show the effectiveness of the proposed algorithm.

**Strengths:**

The strengths of the paper include its originality, quality, and clarity:
1. The idea in this paper is novel to the best of my knowledge. It’s neat to use two samples distanced away from each other to estimate the terms involving two $A$ matrices, which are independent if their gap is large. The sublinear memory requirement also renders this idea a practical approach.
2. The quality of the paper is also a strength. The asymptotic convergence of the proposed algorithm is novel and may bring value to the community.
3. The paper is very well written and easy to follow.

**Weaknesses:**

The paper has weaknesses in its significance and relevant work discussion:
1. The paper may be limited in its significance.
  - On the theory side, the finite time analysis is based on a variant of the proposed algorithm with a projection step, which is absent in the actual algorithm. Thus, the comparison between its convergence rate in this case with that of on-policy TD may not be very valuable. Note that finite sample analysis of the actual algorithm is possible, as also pointed out in the paper. Obtaining such a result can strengthen the paper.
  - On the empirical side, the experiments presented in this paper only focus on Baird’s counterexample and are rather limited. Having more experiments, even in simple environments like FourRoom (Ghiassian & Sutton, 2021; Ghiassian et al., 2024), would help strengthen the claim that the proposed algorithm is effective. In addition to testing the proposed algorithm in environments like FourRoom, a comparison with other GTD algorithms can also make the paper stronger. Other researchers may find it useful to know how the proposed algorithm compares to others in terms of sample efficiency, stability, and hyperparameter sensitivity.
2. The paper also lacks a thorough related work discussion on off-policy policy evaluation (OPPE) with linear function approximation. There have been many follow-up works on the GTD algorithm (Mahadevan et al., 2014; Ghiassian et al., 2020; Yao, 2023). While some of them are cited in the paper, the relationship between these later ideas building on GTD and the proposed method is not thoroughly discussed, which could be useful and inspire future research. Note that Yao (2023) also introduces a GTD variant with one step-size, so it may be necessary to clarify its distinction with your approach. In addition, the paper may also benefit from discussing another line of work that addresses the deadly triad, the ETD family (Sutton et al., 2016; Hallak et al., 2016, 2017; He et al., 2023). Specifically, how does the proposed approach compare to these methods in terms of the optimality of the fixed point and the convergence property? Having a more thorough discussion of these relevant works would strengthen the paper’s positioning.

Ghiassian, S., Patterson, A., Garg, S., Gupta, D., White, A., & White, M. (2020). Gradient temporal-difference learning with regularized corrections. ICML.

Ghiassian, S., & Sutton, R. S. (2021). An empirical comparison of off-policy prediction learning algorithms in the four rooms environment. arXiv preprint arXiv:2109.05110.

Ghiassian, S., Rafiee, B., & Sutton, R. S. (2024). Off-Policy Prediction Learning: An Empirical Study of Online Algorithms. IEEE Transactions on Neural Networks and Learning Systems.

Hallak, A., Tamar, A., Munos, R., & Mannor, S. (2016). Generalized emphatic temporal difference learning: Bias-variance analysis. AAAI.

Hallak, A., & Mannor, S. (2017). Consistent on-line off-policy evaluation. ICML.

He, J., Che, F., Wan, Y., & Mahmood, A. R. (2023). Loosely consistent emphatic temporal-difference learning. UAI.

Mahadevan, S., Liu, B., Thomas, P., Dabney, W., Giguere, S., Jacek, N., ... & Liu, J. (2014). Proximal reinforcement learning: A new theory of sequential decision making in primal-dual spaces. arXiv preprint arXiv:1405.6757.

Sutton, R. S., Mahmood, A. R., & White, M. (2016). An emphatic approach to the problem of off-policy temporal-difference learning. JMLR.

Yao, H. (2023). A new Gradient TD Algorithm with only One Step-size: Convergence Rate Analysis using $ L $-$\lambda $ Smoothness. arXiv preprint arXiv:2307.15892.

**Questions:**

Here are a few questions that might affect the evaluation:
1. Do the choices of $f(t)$ depend on the induced Markov chain’s mixing time? If yes, where is it in the result? If not, why?
2. What is the convergence rate for GTD? Is it the same as the canonical on-policy TD as well?
3. In Line 182, when $f(t)$ was a constant function, what would happen by following the classical convergence results mentioned in Line 186? What’s the consequence of using such a $f(t)$?
4. In Line 227, the paper claims the technique of using a variable interval $T_m$ has not been used in any stochastic approximation and RL literature. Has it been used or studied in other fields? If yes, then it is worth mentioning the relevant literature.
5. In Line 482, the paper claims that the finite sample analysis **confirms** that the proposed algorithm converges reasonably fast, but this analysis is based on a variant of the proposed algorithm with a projection step. Does the efficiency of the variant with the projection step guarantee or generally suggest the efficiency of the base algorithm?
6. Is the proposed algorithm more or less sensitive to the learning rate compared to GTD? Since GTD has two hyperparameters, comparison methods like those in Ghiassian & Sutton, 2021 might be useful here.

---

> ### Author Response · Authors · 2024-11-25
>
> We thank the reviewer for the insightful comments and have integrated all the mentioned works in the new version.
>
> > 1. Do the choices of $f(t)$ depend on the induced Markov chain’s mixing time?
>
> No. It's because we assume the chain mixes geometrically. As long as it's exponential, the exact rate does not really matter because it will be dominated by some other polynomial terms. That being said, the mixing time does affect some constant in the convergence rate.
>
> > 2. What is the convergence rate for GTD? Is it the same as the canonical on-policy TD as well?
>
> Yes. It's $1/t$
>
> > 3. In Line 182, when $f(t)$ was a constant function, what would happen by following the classical convergence results mentioned in Line 186?
>
> If it's constant, we would have $E[A_{t+t_0}^\top A_t]$. But this matrix is not positive definite because it's not $A^\top A$ due to the correlation. So the analysis will not go through.
>
> > 5. Does the efficiency of the variant with the projection step guarantee or generally suggest the efficiency of the base algorithm?
>
> It can only suggest the efficiency of the base algorithm. For example, projected TD and TD have the same rate $1/t$. Projected GTD and GTD also have the same rate $1/t$.
>
> > 6.  Since GTD has two hyperparameters, comparison methods like those in Ghiassian & Sutton, 2021 might be useful here.
>
> We agree with the reviewer but have to leave more empirical study for future work -- 2 weeks is not enough for us to conduct really systematic and rigorous new experiments as those done in Ghiassian & Sutton, 2021.

---

> ### Comment · Reviewer_N6bs · 2024-11-25
>
> Thank you for answering my questions and addressing my comments. Here are some follow up:
>
> > No. It's because we assume the chain mixes geometrically. As long as it's exponential, the exact rate does not really matter because it will be dominated by some other polynomial terms. That being said, the mixing time does affect some constant in the convergence rate.
>
> I'm not entirely sure if the following questions make sense, but can one further exploit the fact that the chain mixes geometrically? How necessary is the chosen example $f(t)$s (in (12) and (13)) or Assumption 4.4?
>
> > It can only suggest the efficiency of the base algorithm. For example, projected TD and TD have the same rate $1/t$. Projected GTD and GTD also have the same rate $1/t$.
>
> Since it can only suggest the efficiency of the base algorithm, I suggest the introduction of the paper be updated to reflect this detail.
>
> > If it's constant, we would have $E[A_{t+t_0}^\top A_t]$. But this matrix is not positive definite because it's not $A^\top A$ due to the correlation. So the analysis will not go through.
>
> I presume you mean the analysis in this paper will not go through as the paper claims that by constructing an augmented Markov chain, one can apply the classical convergence results (Line 186 in the original submission). Then, what results can we expect if we adopt this approach?

---

> > ### Author Response · Authors · 2024-11-25
> >
> > We thanks the reviewer for the prompt reply.
> >
> > > but can one further exploit the fact that the chain mixes geometrically?
> >
> > It might be possible. We envision that if we can set $f(t)$ based on the mixing time, we might be able to get better results (i.e., smaller $f(t)$). However, this approach is not very practical -- $f(t)$ needs to be known to execute the algorithm but the mixing time is in general unknown.
> >
> > > How necessary is the chosen example $f(t)$.
> >
> > This is the best result we have so far without using mixing time in $f(t)$. We have tried smaller $f(t)$ but failed.
> >
> > > Since it can only suggest the efficiency of the base algorithm, I suggest the introduction of the paper be updated to reflect this detail.
> >
> > We totally agree with the reviewer and have updated the introduction and a few places in the main text to avoid overstatement.
> >
> > >  by constructing an augmented Markov chain, one can apply the classical convergence results (Line 186 in the original submission). Then, what results can we expect if we adopt this approach?
> >
> > We are sorry for making this confusion. The classical results typically have two assumptions: the chain needs to be ergodic and the expected update needs to be negative definite. By using a constant $f(t)$, the ergodicity assumption can be fulfilled but the negative definiteness assumption still does not hold. We have revised the submission and added a footnote in page 4 to further clarify this.
> > We thank the reviewer for pointing this out.

---

> > > ### Comment · Reviewer_N6bs · 2024-12-01
> > >
> > > Thank you for your clarifications and for updating the paper. I don’t have any further questions now.
> > > > We agree with the reviewer but have to leave more empirical study for future work -- 2 weeks is not enough for us to conduct really systematic and rigorous new experiments as those done in Ghiassian & Sutton, 2021.
> > >
> > > In addition, I am willing to increase my score if the authors promise to include additional systematic and rigorous experiments to compare the proposed algorithms with other important GTD algorithms, including GTD2 and TDRC. These experiments should provide insights into how the proposed algorithm compares to the alternatives, without necessarily demonstrating that it is the best.

---

> > > > ### Author Response · Authors · 2024-12-02
> > > >
> > > > We thank the reviewer for the reply. And we promise, in next revision, to include systematic and rigorous empirical results following Ghiassian & Sutton (2021) to compare the proposed method with GTD2, TDRC, and target network based approaches as mentioned by the reviewer VCek.

---

> > > > > ### Comment · Reviewer_N6bs · 2024-12-03
> > > > >
> > > > > Thank you for addressing my concerns. As the authors have addressed these points, I have increased my rating on the condition that the additional experiments will be included in the final version.

---

### Official Review · Reviewer_VCek · 2024-11-04

**Soundness:** 3
**Presentation:** 3
**Contribution:** 3
**Rating:** 6
**Confidence:** 3

**Summary:**

The paper aims to develop a convergent algorithm for off-policy temporal difference learning under linear function approximation. The proposed algorithm directly minimizes the L2-norm of expected TD updates (NEU) $||Aw+b||^2$, improving the memory requirement of estimating matrix $A$ from the previous ATD algorithm. Meanwhile, the proposed algorithm reduces the number of learning rates from two to one compared to GTD, another NEU minimization algorithm. It maintains the convergent property with a convergent rate $\tilde{O}(1/t)$. Moreover, the algorithm is tested on Baird’s counterexample and is shown to avoid divergence in the deadly triad.

**Strengths:**

The problem to tackle is well stated, which is to stabilize off-policy learning and improve the previous algorithm ATD and GTD: the proposed algorithm saves memory compared to the ATD algorithm and increases the convergence rate compared to GTD. Also, the paper is clearly written and easy to follow, with rigorously stated assumptions and lemmas.

**Weaknesses:**

The approach needs to be more motivated. GTD is known to suffer from a low convergent rate compared to TD. More experiments to compare the convergence speed and some intuition on why the proposed algorithm can fasten the learning would be great.

Also, the algorithm needs to fit better into the literature. ETD, introduced by Mahmood and colleagues (2015), is another stable off-policy algorithm. Also, a target network is suggested to help convergence (Zhang et al., 2021; Fellows et al., 2023; Che et al., 2024). Che et al. (2024) compare their TD algorithm with GTD on Baird’s counterexample, showing much faster convergence.

Reference

Mahmood, A. R., Yu, H., White, M., & Sutton, R. S. (2015). Emphatic temporal-difference learning. arXiv preprint arXiv:1507.01569.

Zhang, S., Yao, H., & Whiteson, S. (2021, July). Breaking the deadly triad with a target network. In International Conference on Machine Learning (pp. 12621-12631). PMLR.

Fellows, M., Smith, M. J., & Whiteson, S. (2023, July). Why target networks stabilise temporal difference methods. In International Conference on Machine Learning (pp. 9886-9909). PMLR.

Che, F., Xiao, C., Mei, J., Dai, B., Gummadi, R., Ramirez, O. A., ... & Schuurmans, D. (2024). Target Networks and Over-parameterization Stabilize Off-policy Bootstrapping with Function Approximation. arXiv preprint arXiv:2405.21043.

**Questions:**

Is it necessary to take an increasing function f(t)? Besides removing the dependence between data at step t and t+f(t) to establish the convergence proof, is there any other reason to use an increasing function?

---

> ### Author Response · Authors · 2024-11-25
>
> We thank the reviewer for the insightful comments and have integrated all the mentioned works in the new version.
>
> >  is there any other reason to use an increasing function
>
> Yes. If it's constant, we would have $E[ \hat A_{t+t_0}^\top \hat A_t]$. But this matrix is not positive definite because it's not $A^\top A$ due to the correlation. So the analysis will not go through. We added a footnote in page 4 to further clarify this.

---

### Author Response · Authors · 2024-11-25

We thank all the reviewers for providing the insightful comments. We want to provide a response to reviewer N6bs's comment globally first.

> In Line 227, the paper claims the technique of using a variable interval $T_m$ has not been used in any stochastic approximation and RL literature. Has it been used or studied in other fields? If yes, then it is worth mentioning the relevant literature.

We thank the reviewer for this suggestion. After some more careful literature review, we realized that this work is not the first to use a diminishing $T_m$. So we have stepped back from claiming novelty and contribution on this technique in the new version. Instead, we detailed the difference between prior works along the proof in the new version. In short, we argue that our situation is much more complicated than prior works due to that we need to coordinate $T_m$ with both the learning rate $\alpha_t$ and the gap function $f(t)$. By contrast, prior works only need to coordinate $T_m$ with $\alpha_t$. We hope this comment could better clarify the contribution of this work.

---

### Meta-Review · Area_Chair_mfAm · 2024-12-23

**Metareview:**

In this paper, the authors derive a convergent algorithm for off-policy TD learning with linear function approximation. The idea is to use two samples away from each other to address the double sampling issue of GTD. The resulting algorithm directly minimizes the L2-norm of expected TD updates (NEU), thus improves the memory requirement of estimating the term with matrices and reduces the number of learning rates from two to one compared to GTD. The authors prove the asymptotic convergence of their algorithm, and test it on Baird’s counterexample and show that it does not diverge.

The reviewers found the work original, high quality, and easy to read. They found the idea of using two samples distanced away from each other to estimate the term with two matrices novel. They also found memory requirement and smaller number of tunable hyper-parameter (learning rate) desirable in practice.

The reviewers raised some issues, some addressed by the authors during the rebuttals, and some remained that would be good if they are addressed in the final version of the paper.
(-) The paper can benefit from a more thorough discussion of the related work (off-policy policy evaluation with linear function approximation).
(-) Comparing the proposed algorithm with GTD2, TDRC, and target network based approaches.

**Additional Comments On Reviewer Discussion:**

The authors successfully addressed several issues raised by the reviewers during the rebuttals, and they increased their scores. After that all the reviewers are leaning towards acceptance. There are some minor issues that were listed in the meta-review and I hope that the authors will address them in the final version of their work.

---

### Decision · Program_Chairs · 2025-01-22

Accept (Poster)